

# Merits of novel high-resolution estimates and existing long-term estimates of humidity and incident radiation in a complex domain

Helene Birkelund Erlandsen[1,2,3], Lena Merete Tallaksen[2], and Jørn Kristiansen[3]

[1]Norwegian Water Resources and Energy Directorate, Oslo, Norway
[2]University of Oslo, Oslo, Norway
[3]The Norwegian Meteorological Institute, Oslo, Norway

**Correspondence:** Helene Birkelund Erlandsen (hebe@nve.no)

**Abstract.** To provide better and more robust estimates of evaporation and snow-melt in a changing climate, hydrological and ecological modelling practices are shifting towards solving the surface energy balance. In addition to precipitation and near-surface temperature (T2), which often is available at high resolution by national providers, high quality estimates of 2-meter humidity, surface incident shortwave (SW ↓) and longwave (LW ↓) radiation are also required. Novel, gridded estimates of

humidity and incident radiation are constructed using a methodology similar to that used in the development of the WATCH forcing data, however, a national 1x1 km gridded, observation-based T2-data is consulted in the downscaling rather than the 0.5°x0.5° CRU T2 data. The novel dataset, HySN, is archived in Zenodo (https://doi.org/10.5281/zenodo.1970170). The HySN estimates, existing estimates from reanalysis data, post-processed reanalysis data, and VIC-type forcing data are compared with observations from the Norwegian mainland between 1982 and 2000. Humidity measurements from 84 stations are used, and, by

employing quality control routines and including agricultural stations, SW ↓ observations from 10 stations are made available. Meanwhile, only two stations have observations of LW ↓. Vertical gradients, differences when compared at common altitudes, daily correlations, sensitivities to air mass type, and, where possible, trends and geographical gradients in seasonal means are assessed. At individual stations differences in seasonal means from the observations are as large as 7 °C for $T_d$, 62 Wm$^{-2}$ for SW ↓, and 24 Wm$^{-2}$ for LW ↓. Most models overestimate SW ↓, and underestimate LW ↓. Horizontal resolution is not a

predictor of the model's efficiency. Daily correlation is better captured in the products based on newer reanalysis data. Certain model estimates show different dependencies on geographical features, diverging trends, or a different sensitivity to air mass type than the observations.

## 1   Introduction

Geophysical modelling is advancing, and more and more hydrological, ecological, and land surface models (from here on

referred to as land models) are now estimating the surface energy balance (Mueller et al., 2013). Shortwave radiation is the exogenous energy provider to Earth. At middle and higher latitudes, surface downward longwave radiation is an equally important radiative driver at the surface. Estimating the surface energy balance provides a latent heat flux, which in turn can be converted to evaporation or snow melt, key variables for estimating the surface water balance.



Recent studies have showed the added value of using additional forcing data than precipitation and temperature, when modelling evaporation (Milly and Dunne, 2011; Lofgren et al., 2011; Haddeland et al., 2012; Pierce et al., 2013; Stagge et al., 2014) or snow cover (Raleigh et al., 2016; Harpold et al., 2017). High quality and robust diagnoses, forecasts, and projections of evaporation and snow-melt related processes are essential for flood and hydro-power management. Further, gridded data sets

of high quality are needed to statistically bias correct or downscale future climate scenarios (Abatzoglou, 2013), and to spin-up land surface models (e.g. Rodell et al., 2005; Koster et al., 2004; Kristiansen et al., 2012), and to assist model development.

Considerable effort is used to improve process description in environmental models and compare the results of different models. When land models are run without coupling to an atmospheric model, i.e. in offline or stand-alone mode, meteorological near-surface variables, commonly referred to as forcing data, are required. In practice, different communities use different

forcing data estimates, such as the more empirically based estimates from the MTCLIM algorithms (Bristow and Campbell, 1984; Thornton and Running, 1999; Bohn et al., 2013), or estimates from numerical weather and climate models, or a combination of the two (see e.g. Mizukami et al., 2016). The different approaches used makes it more difficult to compare the output across land models, and have resulted in dedicated projects where various models are run with similar forcing data; in e.g. the Inter-Sectoral Impact Model Intercomparison Project, ISI-MIP (Warszawski et al., 2014).

The Norwegian operational hydrological models have historically been calibrated and adapted to use high resolution, gridded 2-meter temperature and precipitation data as forcing data. For this purpose high resolution (1x1 km) data-sets which cover the period from 1957 and until the present with a daily resolution have been developed, namely the SeNorge data (Mohr, 2008; Tveito and Førland, 1999; Lussana et al., 2018b, a). In recent time a gridded, observation-based data set of near surface wind speed has also been developed. Gridded, observation based, high resolution data sets for humidity and and incident radiation

are however lacking.

Previous studies have compared and validated gridded estimates of humidity or incident radiation globally (Bohn et al., 2013; Schmied et al., 2016; Weedon et al., 2011), for regions in USA (Slater, 2016; Mizukami et al., 2014; Pierce et al., 2013; Lapo et al., 2017), coastal Brazil (Almeida and Landsberg, 2003), France (Szczypta et al., 2011), and the pan-Arctic region (Shi et al., 2010). No previous studies have, as far as we know, compared and assessed the quality of high-resolution empirically

based and reanalysis based estimates of humidity and incident radiation for regions within Europe, let alone Norway. This results in an additional and unnecessary source of uncertainty for land modelling in Norway.

Norway's complex topography and coastline may suggest that high resolution data sets would perform better. When land models are run the near surface temperature from the forcing data is usually adjusted to sea level and then to the land model's fine resolution grid cell elevation using a standard atmospheric lapse rate to account for the difference in terrain height in

the forcing data model and the land model to be run. A standard atmospheric temperature lapse rate may be unreasonable in winter time, at high latitudes (Kotlarski et al., 2010; Brinckmann et al., 2016), and in complex terrain (Mizukami et al., 2014); and a much lower resolution in the forcing data grid compared to the land model may increase these error components. Further, if humidity is not also adjusted inconsistency between temperature and humidity will likely result in an unrealistic relative humidity (Haddeland et al., 2006; Weedon et al., 2011). Incident radiation is influenced by variables showing a strong

vertical dependence like near surface temperature and humidity, cloud cover (e.g. Marty, 2000), and local variations in surface



components like vegetation and snow cover (Erlandsen et al., 2017; Rydsaa et al., 2017), and may thus likely benefit from vertical adjustment.

While the spatial correlation may improve in a data set with a high spatial resolution, Decker et al. (2012) highlight the need to address temporal correlation on time scales shorter than monthly in data constructed from reanalyses for the purposes of

forcing land surface models. A high horizontal resolution may lead to a better representation of the average state of a variable, but not necessarily to an improved description of the concurrent temporal evolution of the forcing variables on shorter time scales, e.g. during the rapid passage of a low pressure system with multiple distinct air mass characteristics and precipitation types.

This study addresses the aforementioned sources of uncertainty concerning commonly used estimates of humidity, either

in the form of vapor pressure (VP) or converted to dew point temperature ($T_d$), and incident longwave (LW $\downarrow$), and incident (global) shortwave radiation (SW $\downarrow$) available for long-term land surface modelling in the region by

- construction of an original data set, HySN, to explore the benefit of utilizing a 1x1 km national data set of 2-meter temperature in the post-processing reanalysis data;

- gathering global long-term gridded data sets of humidity and incident radiation from two reanalysis data sets, two post-

processed reanalysis data sets, and two versions of empirically based estimates compiled for continental Norway;

- and aggregating available observations of humidity and incident radiation between 1982-2000 from a variety of providers, and where necessary, implementing quality control routines.

- Further, multiple linear regression models are constructed to provide vertical gradients in both the observations and the model estimates, so that the variables may be adjusted to a similar altitude before their differences are assessed.

- The correlation of model estimates with observations on a daily time scale is explored by compiling anomaly correlation coefficients.

- Finally, the model estimates' cumulative distributions, their sensitivity to weather types, continentality, and latitude, and their decadal trend are compared with the observational data.

Two hypotheses are additionally sought answered: $\mathcal{H}_a$ There are vertical gradients in near surface humidity and incident

radiation in our domain; $\mathcal{H}_b$: The added value of the high horizontal resolution of the more empirically based estimates outweighs the added value of relying on estimates from coarser resolution numerical weather prediction reanalyses.

The data sets considered are two global reanalysis data sets, the NASA Modern-Era Retrospective Analysis for Research and Applications version 2 [MERRA-2] (Bosilovich et al., 2015, 2017), and ECMWFs Era-Interim (Dee et al., 2011), two products based reanalysis data post-processed using higher resolution gridded observational data, the Princeton Global Meteorological

Forcing data set, version 2 (PGMFDv2) (Sheffield et al., 2006) and the WATCH Forcing Data methodology applied to ERA-Interim (WFDEI) (Weedon et al., 2014), and two versions of high resolution empirically based estimates from the pre-processor of the VIC model, a macroscale hydrological model (Liang et al., 1994), largely based on the MTCLIM algorithms. Finally, a





novel data set, the HySN data set, is compiled for the current study and evaluated. HySN is compiled by employing a similar method as used in the development of PGMFDv2 and WFDEI; however in this case Era-Interim near surface humidity and incident radiation is post-processed using a national, high-resolution, gridded 2-meter temperature data set, SeNorge2 (Lussana et al., 2018b).

## 2 The gridded humidity and radiation estimates considered

Long-term data sets that are freely available which can be used to drive hydrological, ecological, and land surface models for the Norwegian domain include the newer re-analyses: MERRA-2 and Era-Interim. Due to computational restrains currently available long-term global reanalysis data have horizontal resolutions ranging from 2°x 2°to 1/2°x 2/3°. The MERRA-2 reanalysis has a resolution of 1/2°latitude x 2/3°longitude. Around Oslo, Norway, this corresponds to a grid cell height and length of about 56 x 42 km. The reanalysis data sets are based on global circulation models ingesting large amounts of observational data by making use of complex assimilation techniques. However, substantial biases may still occur in reanalysis data. Heikkilä et al. (2011) found a mean error of + 42.9 % in precipitation intensity in ERA-40 over Norway between 1961-1990. Bromwich et al. (2016) found a negative bias in Era-Interim surface LW ↓ radiation and precipitation between November 2007 and December 2008 across mid and high latitudes in the Northern Hemisphere

The coarse resolution of reanalysis data and the knowledge of biases which may be present in them has spurred the development of post-processed reanalysis data sets. The PGMFDv2 and WFDEI are data sets consisting of variables relevant for forcing land surface models. The relevant variables are extracted from reanalysis data and post-processed and downscaled with gridded observational data. Both data sets are global and have a horizontal resolutions of 0.5°x 0.5°. PGMFDv2 and WFDEI both adjust reanalysis estimates of humidity and LW ↓ with the gridded, 0.5 x 0.5 degree CRU $T_2$ following the methods described in the development of NLDAS (Cosgrove, 2003). Taking a note from these methods, a novel high resolution product is developed and validated in the current study; Hybrid SeNorge, abbreviated as HySN. HySN is constructed by post-processing Era-Interim humidity and radiances, in a similar manner to PGMFDv2 and WFDEI, but utilizing a national data set, the 1x1 km SeNorge2 $T_2$, rather than the 0.5°x 0.5°CRU $T_2$.

Another source of near surface humidity and incident radiation estimates are the MTCLIM algorithms, which combine first principles from atmospheric physics with empirical extrapolation logic. Precipitation and temperature, variables that often are available from a dense network of surface observation stations, are used to estimate shortwave radiation and humidity. Versions of the MTCLIM routines are used to provide forcing data for a large number of hydrological and ecological models; it has e.g. recently been made available for the Mesoscale Hydrological model (MHm v5.9, doi:10.5281/zenodo.1069202). The variables estimated from MT-CLIM are often utilized for impact studies, e.g. on the impacts of climate change and forest management on ecosystem services in Europe (Bugmann et al., 2017). The algorithms have also been used to generate several gridded data set products of humidity and radiation for the US (e.g. Livneh et al., 2013), and are used to to provide climate change projections for the United States of humidity and radiation (Bureau of Reclamation, 2013). However, several recent studies have found



regionally inconsistent biases in the MTCLIM algorithms (Shi et al., 2010; Bohn et al., 2013; Pierce et al., 2013; Slater, 2016; Mizukami et al., 2014).

The orography and land-masks of the models are presented in Fig. 1. Compared to Era-Interim orography, the SeNorge grid elevation is on average higher (mean: 37 m, median: 13 m, see the red areas in Fig. 1). The difference in maximum elevation is

more than 1000 m. Meanwhile, near the coast and in inland areas the Era-Interim orography is predominantly higher (see the blue areas in Fig. 1). The data sets are summarised in Table 1. Further details considering Era-Interim, MERRA-2, WFDEI, PGMFDv2, WFDEI, HySN, and two data sets from the VIC model's pre-processor, largely based on the MTCLIM algorithms, are presented in the following.

## 2.1   Era-Interim

The Era-Interim (Dee et al., 2011) is a reanalysis data set developed by ECMWF, covering the time period from 1979 until the present. It is is based on a 2006 release of the ECMWF operational model system (IFS Cy31r2) and has a horizontal resolution of about 80 km. It includes a 4-dimensional variational analysis (4D-Var). Surface observations are ingested by optimal interpolation. The variables evaluated in this study are daily means of 2-meter temperature and dew point temperature from analysis times (00, 06, 12 and 18 UTC), and LW ↓ and SW ↓ taken between +12 to +24 hours into the forecast, to allow

for spin-up (see e.g. Weedon et al., 2014).

## 2.2   Modern-Era Retrospective Analysis for Research and Applications 2 (MERRA-2)

MERRA-2 is an atmospheric reanalysis data set developed by NASA, available from 1980 until the present, with a horizontal resolution of 0.5x0.625 degrees (Bosilovich et al., 2015). Mass conservations constraints are imposed so that assimilated observations do not cause large violations to the the global water balance. In MERRA-2 land surface observations are not

assimilated. The data variables used in this study are model orography (Mer, 2015a), pressure and humidity from atmospheric single level diagnostic (Mer, 2015c), LW ↓, and SW ↓ (Mer, 2015b).

## 2.3   Princeton's Global Meteorological Forcing data set version 2 (PGMFDv2)

PGMFDv2 is an updated version of the 0.5x0.5 degree 60 year Princeton's Global Meteorological Forcing data set (Sheffield et al., 2006). The updates are described in Schmied et al. (2016). The humidity, LW ↓, and SW ↓ estimates are based on

the National Centres for Environmental Prediction-National Centre for Atmospheric Research (NCEP-NCAR) reanalysis, but post-processed to comply with the gridded, observation-based time series of precipitation, temperature and cloud cover, with a horizontal resolution of 0.5x0.5, from the Climatic Research Unit (CRU TS 3.2.1), and satellite estimates of LW ↓ and SW ↓.

## 2.4   The WATCH Forcing Data methodology applied to ERA-Interim (WFDEI)

The application of the WATCH Forcing Data methodology to ERA-Interim reanalysis data, WFDEI, is described in Weedon

et al. (2014). The data is available from 1979 until recent time, and has a horizontal resolution of 0.5x0.5 degrees. The humidity,



LW ↓, and SW ↓ estimates are based on Era-Interim data, post-processed to comply with the global gridded, observation-based time series of 2-meter temperature, cloud cover, and inter-annual aerosol loading from CRU TS, using, similarly to PGMFDv2, CRU TS 3.2.1 prior to 2009.

### 2.5 Hybrid SeNorge Era-Interim, HySN (H)

As part of this study, additional estimates of humidity and LW ↓ are derived, using methods based on Cosgrove (2003), adjusting Era-Interim humidity and LW ↓ to comply with the newly developed, 1 x 1 km SeNorge2 $T_2$ data set Lussana et al. (2018b). Further, the Era-Interim SW ↓ estimates are adjusted based on the previously adjusted humidity estimates and the 1 x 1 km orography.

Era-Interim humidity and longwave radiation are vertically adjusted on a daily basis by consulting the daily SeNorge $T_2$. The method differs from that used in the construction of the WATCH and Princeton forcing data, where the reanalysis $T_2$ is adjusted to sea level and then to the CRU grid elevation using a constant lapse rate, before adjusting it on a monthly basis to fit the monthly mean CRU $T_2$. The vertical adjustment of humidity makes use of the common approximation that relative humidity remains constant with height (see e.g. Feld et al., 2013), making it easy to solve for a SeNorge2 compatible dew point temperature $T_d$ based on Era-Interim relative humidity (RH) and SeNorge2 $T_2$. Humidity is corrected to saturation if super-saturation occurs. Surface pressure is adjusted using an approximation of the hypsometric equation. The vertical adjustment of longwave radiation is done by scaling an empirical expression for clear sky LW ↓ to the SeNorge $T_2$ and the previously compiled vertically adjusted humidity estimate. No consistent approach is used in other forcing data sets when vertically adjusting SW ↓. Given that SW ↓ is very sensitive to near surface humidity, and that the Cosgrove (2003) method used above adjusts humidity, we chose to scale the Era-Interim SW ↓ to the ratio of estimated clear sky transmissivity calculated using an empirical equation from Thornton and Running (1999) taking into account the difference in altitude and humidity in the two data sets. A clear-sky type correction approach it thus used both to adjust SW ↓ and LW ↓.

For consistency with SeNorge precipitation and $T_2$ the variables have a temporal resolution of a day, starting from 06 UTC. The data is currently compiled for the time period 1982-2000, and covers the same domain as the SeNorge2 grid. The data compilation is described in detail in the supplemental material. The HySN data product is freely available from Zenodo (https://doi.org/10.5281/zenodo.1970170), and the Python code to generate the data is available on GitHub (https://doi.org/10.5281/zenodo.1435555).

### 2.6 VIC type Forcing Data, VFDv1 and VFDv2

The humidity and radiation estimates referred to here as VIC type Forcing Data (VFD) (see e.g. Bohn et al., 2013; Pierce et al., 2013) are products of the VIC models pre-processor. The VIC model includes the option to generate gridded humidity and radiation from gridded daily precipitation and maximum and minimum temperature. The VIC model pre-processor includes a slightly modified version of the MTCLIM model and algorithms for estimating longwave radiation. The MTCLIM algorithms included in the VIC pre-processors estimate humidity use a modified version of the Magnus-formula with daily minimum temperature used as a proxy for $T_d$ (Kimball et al., 1997). Shortwave radiation is estimated using the Thornton and Running



algorithm (Thornton and Running, 1999). The variables are estimated simultaneously, i.e. the algorithms supply each other with information (Bohn et al., 2013).

Two version of VFD are evaluated in this study. The first version, from here on called VFDv1, uses daily precipitation and mean temperature from SeNorge version 1.1 (Tveito and Førland, 1999; Mohr, 2008), supported by hourly temperature fields from a regional atmospheric reanalysis data set, NOrwegian ReAnalysis (NORA10, Reistad et al. (2011)), with a resolution of about 11 km to compile maximum and minimum temperature using a method similar to Vormoor and Skaugen (2013). The VIC4.0.6 pre-processor is used with default options, i.e. a modified version of MTCLIM4.2, and the TVA clear-sky and Bras full-sky LW ↓ algorithm.

The second version of VIC type forcing data, from here on called VFDv2, is based on slightly different input data, i.e. precipitation, and mean, maximum and minimum temperature from a newer version of the 1 by 1 km SeNorge data, SeNorge2 (Lussana et al., 2018a, b). The VIC4.0.6 pre-processor is used with default options, i.e. a modified version of MTCLIMv4.3, and with LW ↓ estimates based on the Prata (1996) clear-sky algorithm combined with the Deardorff (1978) full-sky algorithm (for further references see Bohn et al. (2013)).

## 3 Study area

Norway is located in the receiving end of the westerlies that pass over the North-Atlantic. This, combined with a long coast lined with mountains provide Norway with 1500 mm of precipitation a year, with distinct regional differences in precipitation amounts received. Although almost 40 % of Noway is covered by forest, evaporation from the land surface is estimated to be less than a fourth of the received precipitation (Hanssen-Bauer et al., 2009). Most of Norway will normally have snow cover the winter season, with the length of the snow season varying from a few days to 300 days a year. Mean temperature (1971-2000) is 1.3 °C, and varies from 7 °C near the coast in southern Norway to -4 °C in the mountains. Between 1976 and 2014 $T_2$ increased with half a degree °C per decade (Hanssen-Bauer et al., 2017).

## 4 Surface observations

The model estimates and observational data are compared for the period 1982-2000. The observational data includes humidity measurements from 84 sites, SW ↓ observed at 10 sites, and LW ↓ observed at two sites. The comparison of the model estimates of incident radiation with stations data is only made possible by including observations from agricultural stations and applying quality control routines. The observations are gathered from the University of Bergen (UiB, SW ↓ and LW ↓ measurements) and from the Norwegian Meteorological Institute's (MET Norway) data repository, which also include measurements from agricultural stations conducted by the institute which is now known as the Norwegian Institute of Bioeconomy Research (NIBIO). The location of the stations used in the comparison with the model estimates are shown in Fig 1.





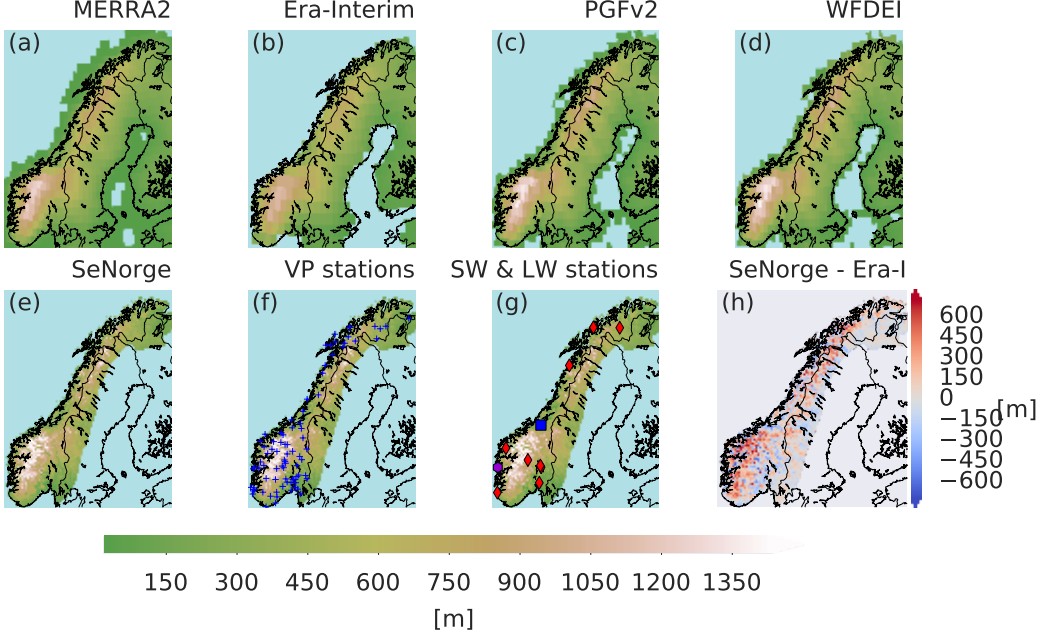

**Figure 1.** The first five images show the orography and land-mask of MERRA2 (a), Era-Interim (b), PGMFDv2 (c), WFDEI (d), and SeNorge (c), respectively, visualized on the SeNorge UTM33 grid with a green-brown color-scheme. For reference, national boarders and the coastline derived from a high resolution data set are delineated in black. The locations of 84 VP station used in the model comparison are denoted with blue crosses in (f). The locations of SW ↓ and LW ↓ stations are marked in (g), with SW ↓ stations are marked with red or purple markers, while LW ↓ stations are marked with a purple (Bergen station) or a blue (Trondheim) marker. The final map plot, (f), displays the difference in meters between the SeNorge and Era-Interim orography in common land areas. Higher elevations in SeNorge are indicated with reds, while blue indicates higher elevations in Era-Interim.

## 4.1 Humidity

Humidity observations from 84 stations are included in the study (see Fig. 1). A minimum of five year's worth of daily data was necessary for the station data to be included, however, most stations have the complete station record (18 years) available. The quality of the observations are fair (pers. comm. Jostein Mamen, MET Norway). The latitude, longitude, altitude, distance to the ocean, the start and end date of the time-series is given, for each station, in the supplement material (Table A.1 and A.2).

## 4.2 Shortwave radiation

The location of the 10 stations included in the evaluation of modelled SW ↓ are displayed with red and purple diamonds in Fig. 1, and cover a latitudinal range from 58.8-69.7°N. Seven of the stations are agricultural stations managed by NIBIO. The latitude, longitude, altitude, distance to the ocean, the start and end date of the time-series, and the percentage of flagged data



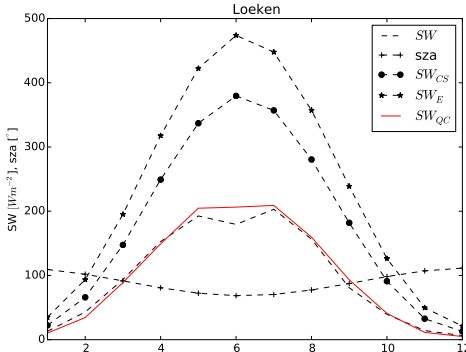

**Figure 2.** Estimated and measured shortwave radiation $\mathrm{Wm}^{-2}$ at Loeken station (61.1°N,9.1°E) for the period January 1991 to December 1999 are depicted. Mean monthly solar zenith angle (sza), (global) shortwave incident radiation at the top of the atmosphere ($SW_E$), modelled clear sky incident shortwave radiation $SW_{CS}$), and station measurements of incident shortwave radiation before (SW) and after ($SW_{QC}$) quality control are shown.

is given, for each station, in the supplement material (Table A.3). The number of days of data used in the validation varies from 5.6 years for Gjengedal to more than 17 years for Bergen.

   Most stations measure global radiation with a Kipp & Zonen CM11 thermoelectric pyranometer. The estimated uncertainty of hourly and daily totals of CM11 may be as low as 3 % in optimal conditions (Grini, 2015). Daily, global SW ↓ measurements
from Bergen station (UiB) are included in the World Radiation Data Centre (WRDC) and are quality controlled by the data provider (UiB, A. Olseth). The daily estimates an uncertainty of 3.5 % Parding et al. (2016). Measurement errors and uncertainty may depend on sensor calibration, placement (e.g. sky-view), the temporal resolution of the measurements, cleaning of the pyranometer, and local weather conditions.

   For stations other than Bergen, quality control procedures conducted this study follows the methodology outlined in Grini
(2015) Table 2. This procedure involves running rtmrun (Godøy, 2013), a Perl wrapper around Libradtran 1.7 (Mayer and Kylling, 2005), a library for radiative heat transfer to provide solar zenith angle (sza), extraterrestrial ($SW_E$) and clear sky ($SW_{CS}$) SW ↓ for each station location, and running the Python-scripts developed in Grini (2015) to screen and flag the data based on automatic quality control tests. Measurements exceeding the upper and lower bounds given in Table 2 were flagged. Additionally, all station time-series were visually inspected at hourly, daily, and monthly form in order to flag erroneous data
not captured by the automatic routines, with emphasis on data-points marked as suspicious due to large hourly increments or very high or low variation in the ratio of observed to extraterrestrial radiation. Fig. 2 shows the mean monthly values of sza, $SW_E$, $SW_{CS}$, the raw, measured values (SW), and values passing the quality control routines ($SW_{QC}$) for Loeken station.

   In the calculation of daily means, values were flagged as erroneous and subsequently excluded from the validation if more than two hourly data points were flagged or missing during daytime. The number of discarded days varied from 4 % at Kise to
29 % at Tromsoe.



### 4.3 Longwave radiation

Bergen station and Voll station (Trondheim), denoted with a purple and blue marker, respectively, in Fig. 1, have observations of incident longwave radiation available for the time period considered. The lack of LW ↓ measurements is not an uncommon challenge (see e.g. Carrer et al., 2012). The stations' latitude, longitude, distance to the coast, and the start date and end date of the data used are listed in the supplement material (Table S.2). Both measurement stations are located within 5 km from the coast (see Fig 1). The measurements from Bergen are managed and quality controlled by UiB. In the first part of the period they are from a Schulze radiation balance meter, while later in the period they are form an Eppley pyrgeometer. The sensors are placed at the roof of UiB. The observation station at Voll, Trondheim was managed by MET Norway from March 1996, until it was shut down. The Trondheim measurements are from a Kipp & Zonen CG 1 pyrgeometer located at the ground. At both stations unshaded sensors were used, possibly leading to slight overestimation due to solar near-infrared radiation contamination (overestimations of 10 % on cloud-free days were found in de Oliveira et al. (2006); Meloni et al. (2012)). The data were quality controlled by visual inspection for spikes and jumps, and by comparing the consistency between the two time-series. The Stefan-Boltzmann black-body longwave radiation was set as an upper limit of the measurements, using the air temperature from the station. If more than two hours were missing or flagged during a day, observations from that day were omitted in the subsequent validation.

### 5 Evaluation methods

Daily estimated values are compared to station observations. The nearest model grid cell is selected from the data sets, without interpolation, to not introduce spatial or temporal smoothing of the meteorological fields (see Hofstra et al., 2010; Gutmann et al., 2012). This study specifically looks into the altitudinal dependence of the humidity and surface incident radiation esti-mates, and as a staring point the estimates without adjustment to the observation stations' altitudes are used.

### 5.1 Vertical adjustment to station altitude

Prior to the comparison of the model estimates with the station observations, the observations and model estimates of VP and SW ↓ were analysed multiple linear regression with geographical features as predictors in order to find vertical gradients so that the model estimates could be adjusted to station altitude. The geographical predictors used were altitude (either the stations' altitude or, for the models, the altitude of the nearest-neighbour grid cell to the stations), latitude above 57∘N, and distance to the coast (calculated in Python using the the Haversine distance from the station to the coastline extracted from a coastline data set (Wessel and Smith, 1996) available via the Matplotlib Basemap Toolkit, implemented at a coarse resolution to not include large inland lakes). The limited amount of SW ↓ measurement stations, and the varying temporal availability of high quality observational data made an evaluation of the altitudinal dependence of the SW ↓ more demanding. The SW ↓ data was first converted to clearness index (CI), which describes the daily incident shortwave radiation fraction of the potential



extraterrestrial radiation at the local position and time (SW $\downarrow$/SW$_E$ $\downarrow$), and daily data from over 1000 concurrent measurements from eight stations were used in the regression.

The model estimates were adjusted to station altitude by multiplying their grid cell values with the difference in altitude to the observation station and a vertical adjustment gradient. For each model the vertical adjustment gradients were computed as the mean of coefficients found for the model in question and those found for the observational data, linearly interpolated from a seasonal to a daily frequency. A similar regression model was constructed to find vertical gradients in LW $\downarrow$ using a well performing data set in lieu of observations due to the limited number of LW $\downarrow$ observation stations.

## 5.2 Evaluation metrics

Seasonality and aggregated means are assessed by plotting the mean monthly station values for the observations and models. The differences between the model estimates and observations at individual stations are displayed in heatmaps. Further, for each variable a table is provided listing several metrics. For each model the mean of the station differences: $\Delta = \overline{\mu_{\text{station,model}} - \mu_{\text{station,observation}}}$, the mean of the absolute station differences: $|\Delta| = \overline{|(\mu_{\text{station,model}} - \mu_{\text{station,observation}})|}$, the largest absolute difference at any station: $|\delta|_{\max} = \max(|\mu_{\text{station,model}} - \mu_{\text{station,observation}}|)$, and the largest absolute difference found at any station in any season: $|\delta^s|_{\max} = \max(|\mu_{\text{season,station,model}} - \mu_{\text{station,station,observation}}|)$ are listed. Also listed are the mean daily anomaly correlation coefficient (ACC, i.e. the daily Pearson correlation coefficient of the time-series where the observed day-of-year mean is subtracted), and the number of stations where the cumulative distribution of daily mean estimates passes (p>0.001) the Kolmogrov-Smirnov test of similarity to the cumulative distribution of the observations. The Kolmogrov-Smirnov test returns the probability of the that the underlying one-dimensional probability distributions are the same ($H_0$).

The similarity between the models and observations on a daily frequency is visualized in Taylor plots, where the normalized standard deviation, the root mean square error, and the correlation coefficient of the de-seasonalized time-series are displayed. The time-series are de-seasonalized by subtracting the observed day-of-year climatology. The correlation coefficient thus corresponds to a non-centred version of the anomaly correlation coefficient (ACC).

## 5.3 Evaluation of geographical gradients

In order to see if the geographical dependencies of the model estimates of humidity and shortwave radiation differs significantly from those seen in the observational data similar multiple linear regression models as those previously constructed to find vertical gradients are used. The predictors are the seasons, altitude (z), latitude above 57°N, and distance to the coast (C). The regression is first performed separately for each model and for the observations. However, a second iteration of regression is performed for each of the models, where the input data is composed of the observational data and model data appended together with the data source denoted. The data source is then used as a categorical predictor allowed to interact with any of the model coefficients. Significance of the interaction term, e.g. between latitude and model source will indicate that the model's latitudinal gradient is significantly different from the gradient seen in the observations.



## 5.4 Air mass type sensitivity

The differences between the model estimates and observations are also inspected for a air mass type dependence. Bower et al. (2007) found significant decreases in the frequencies of dry moderate and dry polar air mass types at Bergen (Flesland) between 1974 and 2000. If the precision of the model estimates are dependent on air mass type the derived changes in the variables with time may be less robust if the frequencies of the air mass types also changes with time. The spatial, synoptic air-mass type classification has been constructed for 48 stations in Europe and seven stations in Norway (Sola, Flesland, Fornebu, Ørlandet, Bodø, Tromsø, and Slettnes) by Bower et al. (see 2007) according to the methods developed in Sheridan (2002) and Kalkstein et al. (1996). The categorization is done by using sub-daily surface observations of temperature, dew point, wind, pressure, and cloud cover at individual stations (often airports).The synoptic weather-typing classifies the local air mass conditions into the categories DP (dry polar), DM (dry moderate), DT (dry tropical), MP (moist polar), MM (moist moderate), MT (moist tropical), and TR (transitional). The dry weather types are associated with clearer conditions, while the moist weather types are associated with clouds and higher humidity. TR days are defined by large shifts in the synoptic variables, i.e. days where the weather type is changing. The MT weather type is often found in the warm sectors of cyclones, while the MP and MM type may be found in the vicinity of a front or in air transported inland from a cool ocean.

## 5.5 Comparison of trends

Where the time-series have a sufficient length and quality the observational data and the model data are inspected for trends. Stations that have less than 10 % missing daily data between January 1985 and December 2000 are considered, and trends are calculated for each calender month using the Mann-Kendal test and by calculating the Sen slope (Hirsch et al., 1982). The analysis is done using the R software and functions within the "trend" package (Pohlert, 2018). 59 humidity stations meet the criteria, and are grouped into five geographical regions, south-west (SW ↓, stations with lat. < 61.8 and lon.<8), south-east (SE, stations with lat. < 61.8 and lon > 8 ), central (C, 61.8 > lat < 64), north-west (NW, lat>64., lon < 20.6 ), and north-east(NE, lat>64, lon>20.6) when assessing trends. For SW ↓ and LW ↓ only the station in Bergen (UiB) meets the criteria. For consistency the humidity observations from Bergen-Florida are inspected for trends as well.

## 6 Comparison between existing long-term estimates and the new hybrid approach, HySN

Daily estimates of near surface humidity and SW ↓ and LW ↓ from MERRA2, Era-Interim, PGMFDv2, WFDEI, VFDv1, VFDv2, and HySN (see Table 1) between 1982 and 2000 are in the following first inspected for vertical gradients in order to adjust the model estimates to station altitude. Further the estimates' quality on a multi-annual time scale is assessed by considering their mean and absolute deviations from station measurements. The model estimates distribution is assessed by comparing their daily cumulative distribution to that of the station observations. The estimates similarity to the observations on a daily time scale is considered by inspecting the anomaly correlation coefficient, i.e. their daily correlation with the measurements after the seasonal cycle has been subtracted, and by considering if their differences to station measurements





show sensitivity to the local daily air mass type, which has been classified for seven Norwegian stations by Bower et al. (2007). For humidity and SW ↓ the geographical gradients in the models are compared with those in the observations using multiple linear regression. A separate sub-section presents modelled and observed trends in each calender month between 1985 and 2000. Humidity trends are compared after grouping the observations and model estimates into five regions, while SW ↓ and

LW ↓ are computed for Bergen, the only location where long-term measurements of SW ↓ and LW ↓ are available within the time-period with little missing data.

## 6.1 Humidity

The model estimates of near-surface humidity are compared against humidity observations from 84 stations. The observation stations used in the validation of humidity are on average located 200 m below the coarse scale grid cell elevation, i.e. rather

in the fjords and on the coast than in the surrounding terrain.

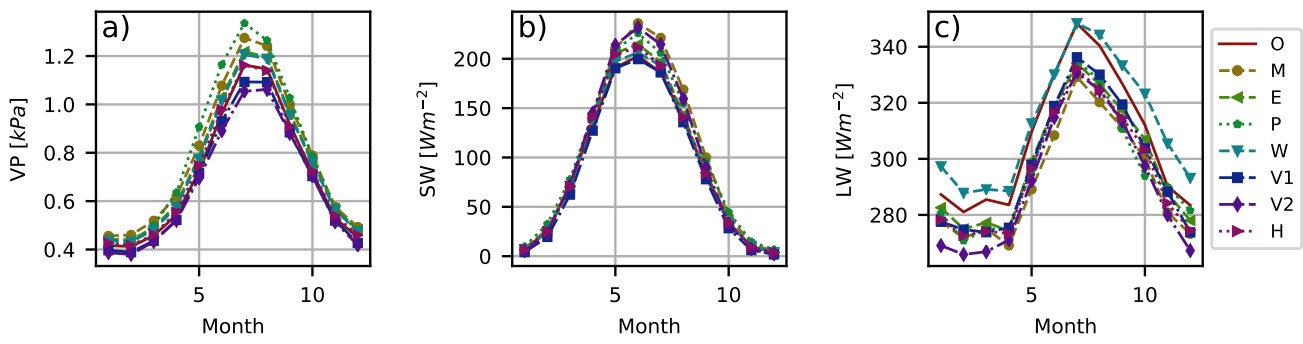

**Figure 3.** The seasonal cycles of monthly 2-m vapor pressure (A), incident shortwave radiation (B), and incident longwave radiation (C), averaged over the locations observations are available, are depicted. The month-of-year is denoted on the horizontal axis.

### 6.1.1 Vertical gradients in humidity

Multiple linear regression of seasonal mean humidity at the location of the humidity stations shows that altitude is a significant predictor of humidity in the observations and all models (not shown). The vertical gradient in the observations is close to the moist adiabatic lapse rate, but varies considerably with season and distance to the coast (C). On average, dew point temperature

decreases with 5.2 °C every kilometre increase in altitude in summer and freeze point temperature decreases with 4.4 °C/km in winter. Regression based on vapour pressure is found to give smaller relative errors than regression based on dew point temperature. This is because dew point temperature has a higher sensitivity to temperature at low temperatures. The observed vertical gradient in vapour pressure is -0.39 hPa/100 m in summer and -0.24 hPa/100 m in winter, and the vertical gradient is weakened with 0.11 hPa/100 m every 100 km away from the coast.



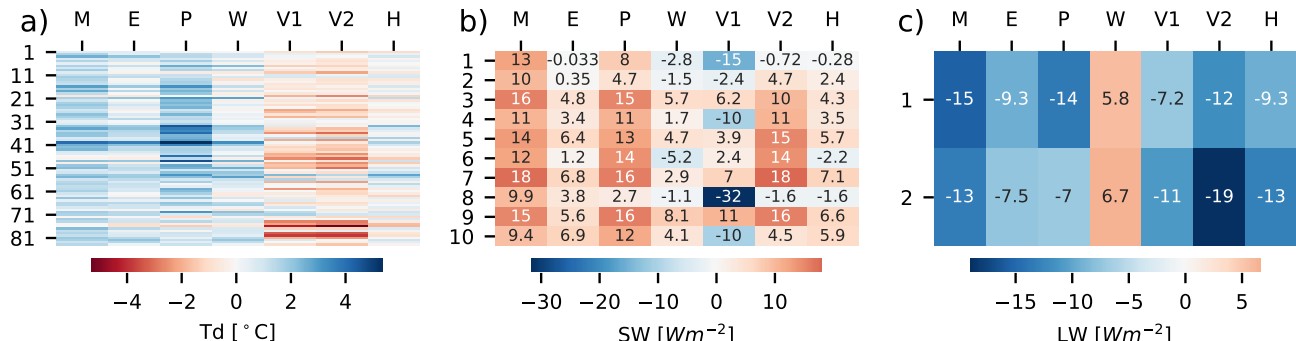

**Figure 4.** For each station, sorted from south to north (y-axis), and each model (x-axis) the differences between the modelled and observed station mean dew point temperature (A), incident shortwave radiation (B), and incident longwave radiation (C) are shown.

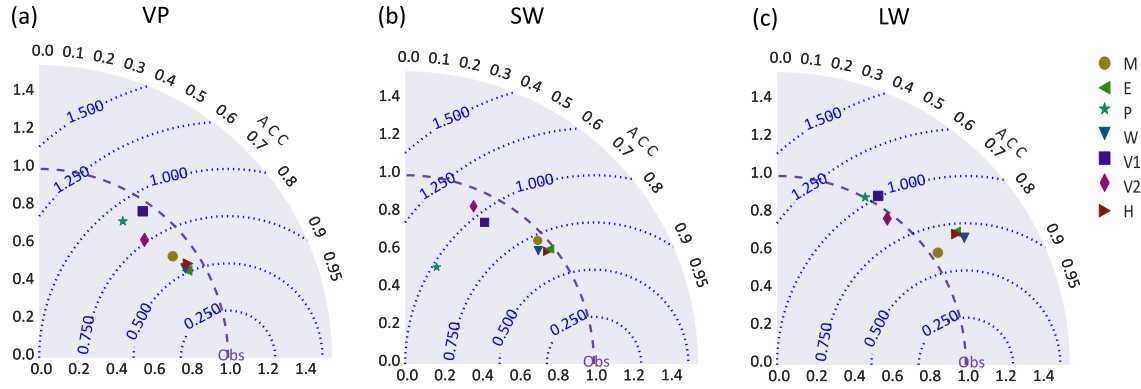

**Figure 5.** Taylor-plots depicting the standard deviation ratio and correlation coefficients (ACCs) for the de-seasonalized time-series.

The vertical gradients in the humidity data differ depending on the data source, e.g. the estimates from PGMFDv2 and WATCH show a weaker decrease with altitude than the observations and other models. For each model the vertical gradients are computed as the mean of the seasonal coefficients found in the regression analyses of the model in question and the observations, linearly interpolated from a seasonal to a daily frequency. The altitudinal adjustment results in a mean difference

5   in humidity, expressed as dew point temperature ($T_d$) of about 1 °C for the coarse scale models, and about 0.06°C for the estimates with a 1 x 1 km resolution. The largest adjustment is an increase in MERRA2's $T_d$ of 7.3 °C at Tafjord station, where the model's orography is 1154 above the station altitude.

### 6.1.2   Differences of humidity estimates to station observations

The seasonal cycle of the observations and models (adjusted to station altitude) is shown in the left plot of Fig. 3. The largest

10   deviations are seen in the period of highest humidity, i.e.during summer. The sign of the average deviations are consistent



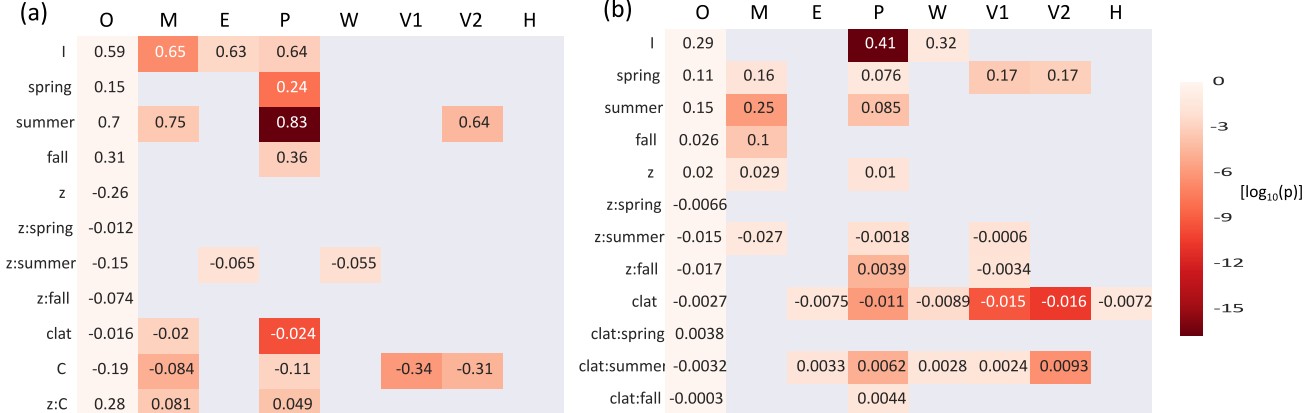

**Figure 6.** Seasonal and geographical dependencies of seasonal humidity (l.h.s.) and daily clearness index (r.h.s.) are depicted. The row-names are the names of the coefficients, including the intercept (I), of the multiple linear regression model. The regression coefficients of the observational data are shown in the leftmost column of each plot, while the coefficients found for the model estimates are only shown if they are significantly different from those of the observations (using a limit of $p<0.01$ for humidity and $p<0.05$ for CI). Lower p-values are indicated with darker colors, using a logarithmic color scale ($\log_{10}(p)$).

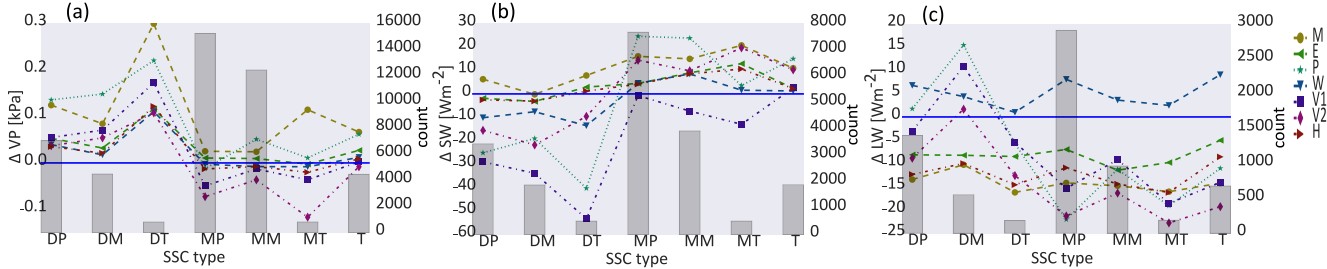

**Figure 7.** Differences between model estimates and observed values binned according to the daily air mass type, classified for nearby stations within Norway (Bower et al., 2007), are depicted. The air mass types are dry polar (DP), dry moderate (DM), dry tropical (DT), moist polar (MP), moist moderate (MM), moist tropical (MT), and transitional (T). The upper plot shows the mean difference in vapor pressure for Saerheim, Aas, Bergen, Bodoe, Tromso, and Maze station. The middle plot shows differences between model estimates and observed incident shortwave radiation at Saerheim (Sola airport), Aas (Fornebu airport), Bergen (Flesland airport), Trondheim (Oerland airport). The lower plot depicts differences in incident longwave radiation binned according to air mass type in Bergen (Flesland airport) and Trondheim (Ørland airport).

throughout the year. PGMFDv2 (denoted with P), MERRA2 (M), and to some degree Era-Interim (E) and WFDEI (W) shows larger estimates than the observations, whereas VFDv1 (V1) & VFDv2 (V2) generally show lower estimates. The HySN (H) estimates follows the mean monthly values of the observations closely. This is also evident in Table 3, where summary of



statics for the humidity estimates are presented, and HySN shows a mean station error in $T_d$ of just $0.1°C$. An aggregated mean similar to the observations does not ensure small deviations from the measurements at individual stations.The VFD estimates have the second smallest deviation in aggregated mean ($\Delta$), however, when considering the average absolute deviation ($|\Delta|$) HySN, WFDEI, and Era-Interim perform better than VFD.

Differences between the model estimates and the station measurements of humidity, expressed as dew point temperature, are are depicted for each station, sorted from south (upper y-axis) to north (lower y-axis) in Fig. 4. The mean absolute difference ($|\Delta|$ or MAE) varies from $0.7°C$ for the HySN estimates to $1.8°C$ for the PGMFDv2 estimates. The largest deviation occurs at an inland station, Fagernes, where PGMFDv2 $T_d$ estimates a $5.4°C$ higher $T_d$ than observed. The figure suggests a latitudinal dependent bias for certain models, and this is further explored in the following part evaluating the models' geographical
gradients.

    The humidity estimates are evaluated on a daily basis by de-seasonalizing the time-series (subtracting the observed day-of-year mean). Fig 5 shows, for each model the de-seasonalized time-series of humidity, the mean temporal correlation coefficient (now equivalent to the anomaly correlation coefficient, ACC), and the mean normalized root mean square error, and mean standard deviation in a Taylor plot. The leftmost Taylor-plot visualizes the mean station metrics for the de-seasonalized time-
series of humidity. The estimates from Era-Interim and and post-processed Era-Interim (HySN and WFDEI) are closest to the observations, and show similar results. MERRA2 also shows a high ACC. PGMFDv2, which is based on an older re-analysis with lower spatial resolution, and the VIC type estimates show slightly poorer results, with ACC ranging between 0.5 and 0.7 (see also Table 3).

### 6.1.3   Evaluation of geographical humidity gradients

Multiple linear regression models are fitted to seasonal mean humidity with the four seasons as categorical predictors, where fall is the baseline season in the model. The geographical predictors considered are altitude ($z$ given in km), latitude (above $57°N$ (lat)), and distance to the coast (C, per 100 km). Further, interaction between altitude and season, and altitude and continentality is included. Each model is paired with the observational data in a common regression model where the data source is included as a categorical predictor.

Fig. 6 displays the regression coefficients for the observations, and the coefficients for the models if they are significantly different ($p<0.01$) from those of the observations. Higher significance is marked with a darker colour. The HySN estimates have similar coefficients to the observations. The regression shows (Fig. 6) that MERRA2 and PGMFDv2 have significantly higher intercepts (higher fall mean values at the coast of southern Norway) than the observations. MERRA2 further shows a stronger latitudinal gradient and a much weaker decrease in humidity with distance from the coast than the observations.
In addition to having a higher intercept than the observations, PGMFDv2 shows a more pronounced seasonal dependency, a weaker continental gradient, and a 50 % stronger latitudinal gradient than the observations. VFDv1 and VFDv2 show a more than 60 % more pronounced decrease in humidity with continentality than the observations. VFDv2 also shows a weaker increase in humidity in summer than the observations.



### 6.1.4   Air mass type sensitivity of humidity deviations

In Fig. 7 (l.h.s) the daily deviations of the humidity estimates are grouped according to the location's daily air mass type classification (SSC-type, see Sec. 5). The classification is available for Sola, Fornebu, Flesland, Bodoe, Tromsoe, and Slettnes station, and the humidity observations considered are from Saerheim, Aas, Bergen, Bodoe, Tromso, and Kirkenes station. All the estimates are too humid in dry weather types. The PGMFDv2 and VFDv2 estimates show considerable overestimations of humidity in dry weather types, and underestimations in moist weather types. The lack of range is consistent with the lower normalized standard deviation seen in the Taylor plot (left plot in Fig. 5). The Era-Interim, WFDEI, and HySN differences also show a slight sensitivity to air mass type, but much less so than the VFDv1, VFDv2, MERRA2, and PGMFDv2.

## 6.2   Incident global shortwave radiation (SW ↓)

SW ↓ observations from 10 sites at the Norwegian mainland are considered. At most locations the coarse scale models' corresponding grid cells have an altitude 300-400 m above station altitudes.

### 6.2.1   Vertical gradients in clearness index (CI)

Multiple linear regression was used to provide a vertical gradients in SW ↓, expressed as clearness index (CI, i.e. the fraction of SW ↓ of the extra-terrestrial incoming radiation ,SW $\downarrow_E$), in order to adjust the estimates to the stations' altitudes. Multiple linear regression including both continentality and altitude as predictors resulted in altitudinal coefficients varying in both magnitude and sign for the different models and observations (not shown). This was likely because the correlation between altitude and continentality varies between 0.56 and 0.86 depending on the data source. Excluding continentality from the predictors provided vertical CI gradients with a consistent sign. The observations show vertical CI gradients of 0.020/100 m in winter, 0.013/100 m in spring, 0.005/100 m in summer and 0.003/100 m in fall (see Fig. 6, r.h.s). The observed SW ↓ increases thus, on average, with altitude in all seasons.

The effect of adjusting the model estimates to station altitude is an average reduction in SW ↓ of 0.7-1.5 $Wm^{-2}$ for the coarse scale models (MERRA-2, Era-Interim, PGMFDv2, and WFDEI) and a reduction of merely 0.1-0.3 $Wm^{-2}$ for the models with a 1 x 1 km grid (VFDv1, VFDv2, HySN). The largest adjustment is a mean reduction of the PGMFDv2 SW ↓ estimate of 4 $Wm^{-2}$ at a station in South-Eastern Norway (Gjengedal).

### 6.2.2   Differences of SW ↓ estimates to station observations

The mean monthly model estimates of SW ↓ averaged over all 10 stations, after adjustment to station altitude, are visualized in the centre plot of Fig. 3. In winter the deviations are small, but in spring and summer all models except VFDv1 overestimate SW ↓. The MERRA2 SW ↓ is on average 35 $Wm^{-2}$ higher than the observations in July, and the VFDv2 SW ↓ is 29 $Wm^{-2}$ higher than the observations in both June and July. Era-Interim, WFDEI, and HySN show the largest overestimations in May, a month when solar radiation high and snow cover is variable.



The centre plot of Fig. 4 depicts the mean difference between the model estimate and the observations of SW ↓ at individual stations. At half of the stations the mean difference between the WFDEI and HySN and the observations are lower than the measurement uncertainty of newer pyranometers in optimal conditions (Sec. 4). The figure further shows that most models consistently overestimate SW ↓. This is not true for VFDv1. While VFDv1 has the second lowest mean monthly deviation

(Fig. 3), its mean absolute difference is larger, on average $10 \, \mathrm{Wm}^{-2}$. This is also evident in Table 4 where summary statics for the SW ↓ estimates are presented.

Table 4 shows that at individual stations seasonal deviations in model estimates from station observations as large as -62 $\mathrm{Wm}^{-2}$. The large underestimation is found in VFDv1 and not VFDv2 at a coastal station in northern Norway, Bodoe. Also listed in the table is the percentage of stations where the daily model estimates, adjusted to station altitude, passes the

10 Komolgrov-Smirnov test of similarity of their cumulative distribution with the observations, which is zero cases for PGMFDv2 and 70 % for HySN.

The similarity of the model estimates to the observations at a daily frequency is visualized in a Taylor plot (Fig. 5, centre). As also seen for the humidity estimates, the PGMFDv2, VFDv1, and VFDv2 have lower ACCs (31-48 %) than the estimates based on newer reanalysis data (73-78 %). Particularly PGMFDv2 shows a considerably smaller variance at a daily frequency

than the observations.

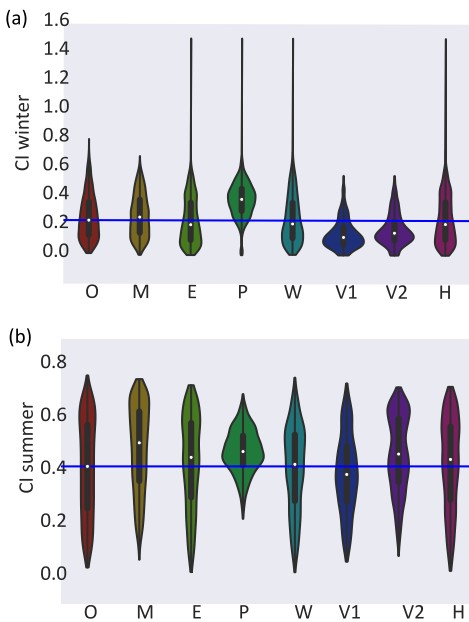

**Figure 8.** Mean daily clearness index (CI) during winter (a) and summer (b) are depicted in violin plots, where the kernel density distribution of the observations and each model is shown, mirrored across the y-axis. The observed median value is drawn in a blue solid line.



### 6.2.3  Evaluation of geographical gradients in clearness index

Similarly as done for humidity, the observations and the corresponding vertically adjusted model estimates of daily CI are compared using multiple linear regression. The seasons, latitude, and altitude are used as predictors, including interaction between season and latitude, and season and altitude. The right plot of Fig. 6 shows the regression coefficients of the observational data in the leftmost column. The coefficients found for the model estimates are only displayed if they are significantly different (p<0.05) from those of the observations. Larger differences ($\log_{10}(\mathrm{p})$) are marked with a darker color.

The estimated intercept of PGMFDv2 stands out in the plot. It is 40 % higher than the estimated intercept of the observations. The second most evident difference is the latitudinal gradient in both VFDv1 and VFDv2, which is several times stronger than observed. PGMFDv2 also shows a stronger latitudinal gradient than observed. Other notable differences are the estimated summer CI values of MERRA-2, which are considerably higher than those seen in the observations.

### 6.2.4  Air mass type sensitivity of SW ↓ deviations

The centre plot of Fig 7 shows the differences between the models' SW ↓ estimates and observations at Aas, Saerheim, Bergen-GFI, Bodoe, Tromsoe station, grouped according to the weather type at nearby weather stations (Fornebu, Sola, Flesland, Bodoe, Tromsoe, respectively). All models except VFDv1 show positive SW ↓ deviations during weather types classified as moist, which occur most frequently. In the less prevalent dry weather types all models except MERRA2 show slightly lower estimates than observed. The largest underestimations are seen for the VFDv1 estimates. Further, the deviations of PGMFDv2, VFDv1, and VFDv2 show a stronger dependency on weather type than MERRA2, ERA-Interim, WFDEI, and HySN. The MERRA2 SW ↓ estimates are, however, overestimated during all weather types. The WFDEI SW ↓ estimates show considerable deviations from the Era-Interim estimates, with larger underestimations found during both dry polar and dry tropical weather types, and considerably lower overestimations found on days classified with a moist tropical weather type. Grouping the clearness index into either dry or moist/transitional weather types shows that the observed CI decreases on average with 0.22 in moist/transitional types. A similar decrease is seen in MERRA2, Era-Interim, WFDEI and HySN, but not in VFDv1 and VFDv2 (-0.12), or PGMFDv2 (-0.05). The summer and winter distributions of clearness index at the stations considered is depicted in Fig 8. It is evident that PGMFDv2 spans a much smaller range of transmissivity than observed, in both summer and winter, and that the VIC type estimates have a bias towards low estimates and show less variability than observed in winter.

Since both WFDEI and HySN are based on Era-Interim, and Era-interim show overestimations of SW ↓ in summer, where observations were available, the differences in estimated SW ↓ in Era-Interim and the observations were inspected for dependencies on differences in modelled and observed cloud cover, near surface humidity and snow cover using regression. The results varied with both season and location, but for the aggregated data significant dependencies on differences in observed and modelled 2-m humidity and snow cover were found (with higher snow cover in the model associated with higher SW ↓ estimates in the model), and also, in the warm season larger overestimations were seen in Era-interim when the model produced high clouds.





### 6.3 Incident longwave radiation (LW ↓)

Only two stations have LW ↓ observations available during the validation period, Bergen-GFI (western Norway), and Trondheim-Voll (central Norway); and both are located near the coast (see Fig 1 and the supplement material). More than 17 years of daily measurements are avaliable from the Bergen-GFI station, while at Trondheim-Voll only about two years of observations are available.

#### 6.3.1 Vertical gradients in LW ↓

Since only two stations have longwave observations, no altitudinal gradient can be inferred from the observations, instead altitudinal gradients are taken from Era-Interim. The previous comparison of altitudinal gradients within the observations and models has shown that Era-Interim has similar altitudinal gradients to the observations. Era-interim has further not previously been vertically adjusted. The vertical gradients found ranged from -4.0 $\mathrm{Wm}^{-2}/100\mathrm{m}$ in December to -0.6 $\mathrm{Wm}^{-2}/100\mathrm{m}$ in June, and were weakened with 0.20 $\mathrm{Wm}^{-2}/100\mathrm{m}$ for every 10 kilometres away from the coast, and strengthened with 0.23 $\mathrm{Wm}^{-2}/100\mathrm{m}$ for every latitude North of 57°N. On average the vertical gradients were around -4.5 $\mathrm{Wm}^{-2}/100\mathrm{m}$ in winter, and -1.8 $\mathrm{Wm}^{-2}/100\mathrm{m}$ in summer. The gradients were temporally interpolated to day-of-year values and applied to the models in order to adjust the estimates to the altitude of the two observational station. The largest change in the estimates due to altitudinal adjustment is an average increase of 8.6 $\mathrm{Wm}^{-2}$ for MERRA2 when adjusted to the 278 m lower altitude of Bergen station compared to MERRA2's grid cell altitude.

#### 6.3.2 Differences of the LW ↓ estimates to station observations

The right plot of Fig. 3 depicts the mean monthly LW ↓ at the two stations. Summary statics for the LW ↓ estimates after adjustment to station altitude, are also presented in Table 5. At the two stations all models except WFDEI estimate lower values than observed in all months. WFDEI, however, estimates on average 10-15 $\mathrm{Wm}^{-2}$ more LW ↓ than observed from October trough January. The largest absolute differences are found in MERRA2, where LW ↓ is underestimated with 8 $\mathrm{Wm}^{-2}$ in winter and with 25 $\mathrm{Wm}^{-2}$ mid-summer. The remaining models estimate 11 to 17 $\mathrm{Wm}^{-2}$ less LW ↓ than observed in summer, and show smaller underestimations in winter.

The skill of the model estimates in capturing the day to day variability in LW ↓ is visualized in the rightmost Taylor plot of Fig. 5, indicating the correlation and normalized standard deviation of the de-seasonalized time-series. The estimates based on newer reanalysis data, MERRA-2, Era-Interim, WFDEI, and HySN have anomaly correlation coefficients around 0.8, while PGMFDv2, VFDv1, and VFDv2 have lower ACCs (46-60 %).

#### 6.3.3 Air mass type sensitivity of LW ↓ deviations

The lower plot in Fig. 7 depicts the differences between the model's LW ↓ estimates and the observed values at Bergen and Trondheim station, grouped according to the daily weather type (classified at Flesland and Ørlandet station). On days classified with moist or transitional weather types all models except WFDEI underestimate LW ↓. PGMFDv2, VFDv1, and



VFDv2 clearly have weather type dependent deviations from the observations, with underestimations in moist weather types, and smaller underestimations or even overestimations compared to the observations in dry weather types. The MERRA2, Era-Interim, WFDEI, and HySN estimates largely show similar differences to the observed values in all weather types. An additional comparison of the Era-Interim estimates at Bergen, where cloud observations are available, showed no difference
in average deviation in the estimates of incident longwave radiation on common clear-sky days compared to the remaining days (not shown). The lower estimates of incident longwave radiation in Era-Interim is thus likely not primarily related to differences in cloud properties.

### 6.4 Modelled and observed trends in near surface humidity, SW ↓, and LW ↓ between 1985 and 2000

January 1985 is considered the start of the brightening period in Europe after a period of SW ↓ dimming period due to aerosol
emissions (see e.g. Wild et al., 2005). In the following, where available, observations and co-located model estimates are inspected for trends in near surface humidity, SW ↓, and LW ↓ between January 1985 to December 1999.

After screening the observational time-series 59 humidity stations are grouped into five geographical regions (south-west (SW ↓), south-east (SE), central (C), north-west (NW), and north-east(NE), see Sec. 5). Table 6 lists the results of the trend tests for each calender month and region, listing only the Sen slope if the Mann-Kendall trend test is significant at a 5 or 10 %
level, with the latter denoted in *italics*. In all regions except the north-eastern part of Norway significant increases in observed $T_d$ occurred in September between 1985 and 2000. The observations further show an increase in $T_d$ in April in South Norway, in July in the north-eastern part of Norway, and a decrease in May $T_d$ in central Norway. For South and central Norway all the models capture the increase in September $T_d$. The models reproduce the observed trends in spring and in northern Norway to a lesser degree.

Bergen is the only location in Norway where long-term records of incident shortwave or longwave radiation are available with few missing data for the period 1985-2000. Between January 1985 and December 1999, observations from the University in Bergen show trends in annual SW ↓ of 1.7 $\mathrm{Wm^{-2}}$/decade (p<0.1), while LW ↓ decreases with -8.4 $\mathrm{Wm^{-2}}$/decade (p<0.001). At the nearly co-located measurement station of the Norwegian Meteorological Institute, at Bergen-Florida, annual dew point temperature has a trend of 1.2 $^\circ C$/decade (p<0.0001).

In individual calendar months larger trends were found. Observed August mean SW ↓ increased with 51 $\mathrm{Wm^{-2}}$/decade (Table 8). The observations also show significant, lesser increases in October and December. Apart from WFDEI and VFDv1, which show no significant SW ↓ trends, the models reproduce a significant increase in SW ↓ in August, and no models show trends of the opposite sign during the time-period considered.

Monthly mean LW ↓ in Bergen shows a significant decrease in several months of the year, the largest is found in May,
with -21 $\mathrm{Wm^{-2}}$/decade. Also in August, October and December, months when concurrent increases in shortwave radiation were found (Table 8), the observations show significant decreases in incident longwave radiation. None of the models show equally strong negative trends in monthly mean LW ↓ in their corresponding grid cells. MERRA2 and Era-Interim show no significant trends during the time-period, whereas PGMFDv2, WFDEI, and HySN exhibit one single month with a negative trend. Meanwhile, VFDv1 and VFDv2 show weakly positive trends in September. The increasing trend in incident longwave



radiation in September in the VIC Type estimates may be related to the concurrent increase in humidity. All the models also show an increasing trend in humidity in the grid cell covering Bergen in September, however, the models generally show weaker trends than observed at Bergen-Florida station (see Table 7).

## 7  Discussion

Historical estimates of humidity and incident shortwave and longwave radiation have been compared to station observations from mainland Norway between 1982 and 2000. 84 stations provide vapor pressure (VP) observations, 9 stations SW ↓ observations, while only two stations observed LW ↓. The estimates evaluated are from two reanalysis data sets, MERRA2 and Era-Interim, three data sets composed of reanalysis data blended with gridded observational data, PGMFDv2, WFDEI, and HySN, and two versions of the VIC type forcing data, estimates based on gridded observational data combined with empirical

algorithms.

Differences between the estimates and observations are not necessarily due to errors in the estimates, as a vertical adjustment to station altitude does not suffice to require that the model grid cell estimates should equate to the observed values. The numerical model estimates may still differ from the observations for valid reasons, such as differences in snow cover, differences in land cover type (the observations are from sensors usually located over grass or, in some cases, on top of buildings), and due

to the averaging out of sub-grid variability in the models (see e.g. Göber et al., 2008). The uncertainty in the observations may also contribute to the differences. However, large differences may however suggest biases in the estimates.

### 7.1  Vertical gradients

Significant vertical gradients were found for humidity, incident shortwave, and incident longwave radiation, justifying an altitudinal adjustment to station altitude before comparison of the model estimates with the station observations ($\mathcal{H}_a$). The

altitudinal vapour pressure gradients found here were on average -0.25 hPa/100 m in winter and -0.34 hPa/100 m in summer. The summer gradient is similar to what Marty (2000) found in the Alps in summer, -0.35 hPa/100 m; however, the winter gradient is considerably higher found in the Alps (-0.14 hPa/100 m). The impact of adjustment to station height was small for the estimates with a finer spatial scale, on average just a 0.06°C change in $T_d$, while for the coarser scale estimates, MERRA2, Era-Interim, PGMFDv2, and WFDEI, the impact of the vertical adjustment was considerably larger, resulting in an average 1°C

increase in $T_d$. The WFDEI and PGMFDv2 showed weaker vertical humidity gradients than observed. This may be a result of the interpolation techniques employed in the CRU $T_2$ data set used to bias correct and downscale both the Era-Interim and NCEP-NCAR re-analysis, or due to the use of a constant temperature lapse rate of 6.5° C when interpolating the temperature of the two re-analyses to the CRU orography. Notably, the vertical gradients in near surface humidity in MERRA2, a reanalysis where surface observations are not assimilated (see Table 1), are similar to the vertical gradients found in the observations and those found in Era-Interim.

those found in Era-Interim.

Observed SW ↓ in the form of clearness index (CI, see Sec. 5) showed the highest altitudinal gradient in winter, a slightly lower gradient in spring, and rather low gradient in summer and fall. The vertical gradients found are larger than the gradient





of 0.00295 /100 m used in the implementation of the Bristow and Campbell (1984) algorithm in historical versions of the VIC pre-processor (MT-CLIM versions before 4.2, before the Thornton and Running (1999) algorithm was implemented). Though the CI-gradient is stronger in winter, the considerably smaller amount of SW ↓ received leads to a weaker gradient in SW ↓. The CI-gradients translates to SW ↓-gradients of about 0.3 $\mathrm{Wm}^{-2}$/100m in fall and winter, 1.6 $\mathrm{Wm}^{-2}$/100m in spring, and 1.2 $\mathrm{Wm}^{-2}$/100m in summer. Marty (2000) found all-sky gradients in SW ↓ in the Alps of 1.1 $\mathrm{Wm}^{-2}$/100m in winter and 0.7 $\mathrm{Wm}^{-2}$/100m in summer. The differences between the gradients found here and those given in Marty (2000) may likely be explained by the differences in the received extra-terrestrial radiation, and differences cloud and snow cover climatology. The models largely showed similar vertical CI-gradients to the observations. The exceptions were PGFMDv2 and VFDv1; PGMFDv2 showed significantly (p<0.01) weaker vertical gradients with a weaker seasonality, and VFDv1 produced a stronger vertical CI-gradient in summer than in winter. The adjustment of the coarser scale estimates resulted on average to a five times larger change in SW ↓ for the coarse scale estimates than the finer scale estimates. The regression-based vertical adjustment produced similar SW ↓ estimates for Era-Interim and the HySN estimates.

Since two LW ↓-stations located more than 400 km apart could not provide an observation-based vertical gradient in LW ↓, Era-Interim was consulted instead. The gradients were, on average, -4.5 $\mathrm{Wm}^{-2}$/100m in winter, and -1.8 $\mathrm{Wm}^{-2}$/100m in summer. Marty (2000) found vertical gradients in incident longwave radiation of -2.8 $\mathrm{Wm}^{-2}$/100m in winter and -3.1 $\mathrm{Wm}^{-2}$/100m in summer for the Alps, and of -4.1$\mathrm{Wm}^{-2}$/100m in winter, and -2.6 $\mathrm{Wm}^{-2}$/100m in summer when considering a subset of observation stations in Switzerland. The different vertical gradients found may be explained by differences in temperature and humidity gradients, different climatological distributions of clouds, and the difference in initial temperature, as LW ↓ is a function of temperature to the power of four. The regression based vertical adjustment of Era-Interim LW ↓ estimates resulted in a larger correction of LW ↓ than the clear-sky-type adjustment implemented in HySN, alluding to that the clear-sky-type altitudinal adjustment implemented in similar data products might be too low, especially for locations with a maritime climate, like Bergen.

## 7.2 Differences to station observations

### 7.2.1 Humidity estimates

The empirically based model estimates, VFDv1 and VFDv2, show on average slightly lower estimates of humidity than observed. Both VFD-type estimates are found to show a 50 % stronger decrease of humidity with continentality than the observations (see Sec. 6.1.3). The modified version of the Magnus-type formula based on Kimball et al. (1997) used in MT-CLIM to generate the VFD humidity estimates is likely not appropriate for Norway. Previous studies, e.g. in the development of gridded climate variables by New et al. (1999), and in the application of the MT-CLIM model over complex terrain in Australia (Thornton et al., 2000), and in the western US (Pierce et al., 2013) found that the Kimball et al. (1997)-method did not result in overall improved humidity estimates. Indeed, in Kimball et al. (1997) the method is found to give improved estimates of humidity in locations where the ratio of potential evaporation to annual precipitation is larger than 2.5. In most regions of Norway this ratio is well below unity. The more conventional method of using daily minimum temperature as a proxy for dew point temperature





will likely give relatively smaller overestimations of humidity than the underestimations resulting from using the Kimball et al. (1997)-method.

The reanalysis and reanalysis-based estimates all overestimate humidity, and the overestimations are generally higher in weather types classified as dry according to the methodology of Bower et al. (2007). Particularly MERRA2 and PGFMDv2
overestimate humidity in dry conditions. The same two models also show a significantly stronger decrease in humidity with latitude than observed. MERRA2 also shows a weaker decrease of humidity with continentality. The weaker decrease of humidity with continentality seen in MERRA2 may perhaps be partly explained by the model's coarse resolution and land mask (see Fig. 1), and MERRA2's exaggerated latitudinal gradient in humidity in Norway may perhaps be associated with MERRA2's larger latitudinal gradient in SW ↓.

The humidity estimates from HySN match the observations best, considering all metrics except from the anomaly correlation coefficient (ACC). The ACC of the Era-Interim estimates are marginally higher (0.02) than in HySN. This is likely due to the capping of relative humidity at 100 % when applying the SeNorge2 temperature in the development of HySN. Combining the methods outlined in Cosgrove (2003), which for humidity relies on the assumption of constant relative humidity with altitude, a high-quality reanalysis data set (Era-Interim), and a high-resolution, national, observation-based temperature data set is found
to provide high quality daily estimates of humidity in the current study region. The coarser, reanalysis-based data sets generally show higher ACCs than the VFD estimates. Numerical weather predictions (NWP) models are skilled at capturing synoptic events, i.e. weather or climatological patterns on a spatial order of 1000 kilometres, and a temporal order of days or weeks, such as cold air outbreaks and the changing sources of air-masses during the passage of warm and cold fronts. Though the NWPs may have systematic biases and a much lower spatial resolution than empirically based estimates, it is not surprising
that they are skillful in representing daily weather variability.

### 7.2.2    Incident shortwave radiation

Shortwave incident radiation is on average overestimated for all model estimates except for VFDv1. HySN, Era-Interim, and WFDEI vary in obtaining the highest ranking depending on the metric considered. For instance, WFDEI shows a slightly lower average deviation from the observations than Era-Interim and HySN. On the other hand, WFDEI shows larger underestimations
in dry weather types than Era-Interim and HySN (Fig 7). Overall, the three models provide vertically adjusted estimates of incident SW ↓ close to the observations, with average deviations from station measurements below $4\,\mathrm{Wm^{-2}}$, and ACCs above 0.76.

The average difference between the Era-Interim estimates and the observations is smaller than in Urraca et al. (2018), where an average overestimation of $12\,\mathrm{Wm^{-2}}$ was found when comparing Era-Interim SW ↓ estimates to station measurements
in Europe between 2010-2014. The smaller difference seen in the current study may in part be explained by the relatively smaller amount of solar radiation reaching Norway, the different time periods considered, and the vertical adjustment included in the current study. Urraca et al. (2018) also found that MERRA-2 shows poorer results than Era-interim; with average overestimations of $18\,\mathrm{Wm^{-2}}$. This is consistent with our findings; where MERRA-2 has the highest mean deviation from the observations of any of the considered estimates. Overestimations of incident shortwave radiation over land is not only an issue



of reanalysis data sets covering Europe, but has been a long standing issue in global (Wild et al., 2015) and regional climate models (Katragkou et al., 2015; Jerez et al., 2015).

Two versions of VIC style Forcing Data are evaluated in the current study. The two versions differ in their input data and in the version of VIC pre-processor used. The oldest version of the VFD-data sets is partly based on a 11 km national reanalysis (NORA10) to provide maximum and minimum temperature. The older version showed large underestimations of incident shortwave radiation at several stations, particularly near the coast in Northern Norway (Fig. 4). These findings are in line with Bohn et al. (2013), where the MT-CLIM algorithms were found to underestimate incident SW ↓ radiation with 26 %, on average, at coastal sites. The MT-CLIM algorithms implemented in VFD rely in part on the diurnal temperature range to estimate cloud cover, using a low range as an indication of cloud cover. Near the coast the diurnal temperature range may be low due to the moderating influence of the nearby ocean, due to its high heat capacity. The more recently compiled version of VFD data, VFDv2, which is based on a newly developed, high resolution gridded data set of $T_{min}$ and $T_{max}$, does not show similar underestimations near the coast of Northern Norway. The different estimates produced points to that great care must be used to make sure the VIC style Forcing Data have consistent input data and algorithm-versions if the data is used e.g. in climate change impact studies.

The newer and higher resolution input data used in VFDv2 did not result in a lower mean absolute station deviation, as its SW ↓ estimates were consistently overestimated. Both VFD-versions show a much stronger latitudinal gradient than observed and a too strong altitudinal gradient in summer. The latter finding is in line with Mizukami et al. (2014), where VFD-type estimates for the Colorado River basin showed increasing overestimations of SW ↓ with increasing altitude. The exaggerated latitudinal gradient in SW ↓ may be connected to the use of the diurnal temperature range in the algorithm. Bohn et al. (2013) found that the relationship between cloud cover and the diurnal temperature range breaks down for ranges below $5°C$. Further, New et al. (1999) states that the relationship between diurnal temperature range and cloud cover is weak at around 60°N in winter, and further becomes positive in the Arctic.

Binning the estimates according to air mass type shows that the PGFv2 and VIC Type estimates show less sensitivity too the prevailing weather type than the observations. The observations and the remaining model estimates show a decrease in clearness index of about 0.22 on days classified with moist or transitional weather types rather than dry, while the VIC Type estimates and PGFv2 show reductions of 0.12, and 0.05, respectively. On average, the VIC Type estimates and PGFv2 underestimate incident radiation in dry weather types (see Fig. 7). The similarity between the PGMFDv2 and VFD estimates of SW ↓ may be explained by the fact that the PGMFD SW ↓ is bias corrected based on gridded cloud cover from CRU using the Thornton and Running (1999) relationship between SW ↓ and cloud cover, which is also used in VFD. Further, the gridded CRU cloud cover data set is a secondary or derived observational data set, which is, similarly to VFD, in part based on regression using diurnal temperature range as a predictor (New et al., 1999). The lower sensitivity to air mass type found in PGMFDv2 and the VIC Type Forcing Data might contribute to the lower ACC found for these estimates.



### 7.2.3   Incident longwave radiation

The evaluation of incident longwave radiation is compromised by the lack of observational data. Only two sites observe incident longwave radiation in the considered time-period. The difference between the annual mean of the model estimates and observations are, for the two stations considered, larger for incident longwave radiation than for incident shortwave radiation. The annual deviations ranges from -16 to +7 $\mathrm{Wm}^{-2}$. Underestimations of monthly means are found throughout the year for all models except WFDEI. The deviations from the station observations are for WFDEI, MERRA2, Era-Interim, and HySN largely similar in all weather types, i.e. the underestimations are also found on days classified with dry weather types. An additional evaluation of the Era-Interim LW estimates for Bergen, where cloud observations are available, showed that the deviations from station observations were similar on days where clouds were present both in observations and the model and the remaining days.

While overestimation of incident shortwave radiation has been a long standing issue in many climate models and reanalyses, incident longwave radiation is typically underestimated (Katragkou et al., 2015; Li et al., 2016; Zib et al., 2012; Wild et al., 2017). The causes of the underestimation are however debated. Li et al. (2016) points to an improper representation of inter-action between radiation and suspended frozen water particles in the atmosphere (solid hydrometeors) as a culprit, while Zib et al. (2012) speculates that errors in simulated aerosols, water vapor content, and cloud properties (rather than cloud amounts) are the cause. Local issues such as longwave emissions form nearby terrain may also contribute to the deviations (Rontu et al., 2016). Lastly, observational uncertainty confounds the picture further, particularly given that the two sensors were unshaded.

The anomaly correlation coefficients are, as also seen for humidity and incident shortwave radiation, considerably lower for PGFMDv2 and the VFD estimates than the estimates based on newer reanalysis data (0.46-0.60 vs 0.79-0.82). This may be caused by the representation of clouds in the models. As discussed for the PGFMDv2 and the VFD SW ↓ estimates, the use of diurnal temperature range as a proxy for cloud cover may not be suitable for the current maritime, high latitude study region.

### 7.3   Trends

The analysis of observed humidity trends between 1985 and 2000 showed significant increases in April in South Norway, a decrease in May in central regions of Norway, significant increases in July in the North-Eastern part of Norway, and increases in all regions except North-Eastern part of Norway in September. All the data-sets, both the reanalysis-based estimates and the more empirically based VFD estimates, capture the increase in humidity seen in September. The significant increases found in humidity when averaging over the stations in South-East and South-West Norway in April are not seen in any of the models. The VFD estimates do, however, capture some of the increase in humidity which was seen in the measurements from Bergen Florida (Table 7). A recent study by Nilsen et al. (2017) found that changes in large scale weather patterns can, in part, explain significant increases in 2-m temperature between 1981 and 2010 seen in Scandinavia in September, but not explain most of the increase seen in April. Another inquiry by Rizzi et al. (2017) found that the increasing temperature trends seen in May in many parts of Norway showed a strong correlation with a concurrent decrease in snow cover. The decline in snow cover in May found in Rizzi et al. (2017) was particularly strong in low lying areas. If the changes in temperature and humidity are





connected to local changes in snow cover it is possible that the coarser scale reanalysis data, which often have a mean grid cell altitude above the measurement station elevation, do not capture the measured changes.

Surface incident radiation was inspected for trends between 1985 and 2000 at the one station where measurements are available in the time-period with little missing data, Bergen. A hardly significant (p<0.1) annual trend in SW ↓ was found in the observations, 1.7 Wm$^{-2}$/decade. However, in individual calendar months larger trends were found. The largest trend, 51 Wm$^{-2}$/decade was found in the observations in May. The observed August trend was reproduced fairly well in Era-Interim, PGMFDv2, and HySN, and a weaker, but still significant trend was seen in MERRA2 and VFDv2. While Era-Interim largely reproduces the trend, WFDEI show no significant trends. For the considered location the post-processing of Era-Interim radiances based on CRU-cloud cover and inter-annual aerosol loading conducting in the production of WFDEI has an negative effect on its ability to reproduce of the observed trend. The clear-sky type post-processing of Era-Interim implemented in HySN-estimates left the trend close to its original value. The two versions of VFD also differed in their ability to capture the SW ↓-trend, which might to do with the maritime location of Bergen and VFDv1 coarser input data for $T_{min}$ and $T_{max}$. A previous study by Parding et al. (2016) showed that circulation type changes could account for a large part of the dimming that was observed in Bergen before around 1980, but a lesser fraction of the subsequent brightening. The fact that Era-interim, which does not explicitly account for inter-annual aerosol-changes, picks up the trend, when WFDEI, where a correction for inter-annual aerosol-loading has been applied, does not implies that a considerable part of the trend before 2000 must be included in the indirect effects of aerosol changes, which Era-interim assimilates, or other factors. On the other hand, MERRA-2 accounts for inter-annual aerosol-loading in the time-period considered and captures a positive, albeit weaker trend in SW.

The annual trends in LW ↓ during the same period in Bergen were larger in magnitude than those found for SW ↓, -8.4 Wm$^{-2}$/decade. The observed trend in any calender month was larger for SW ↓, while the LW ↓ trend showed more consistency throughout the year. More of the models reproduced the SW ↓-trend than the LW ↓-trend. Both versions of the VIC Type forcing data, VFDv1 and VFDv2, simulated an weak increasing trend in September. Given that the VFD-estimates did not produce changes in SW ↓ in the same month, the increase is likely due to the clear-sky parametrization and the concurrent simulated increase in September $T_d$. WFDEI and HySN reproduce the decrease in LW ↓ seen in August, while Era-Interim does not. This points to that changes in near-surface temperature, which is used as a scaling factor and to adjust near surface humidity in WFDEI and HySN, captures the signal which contributes to the decrease in LW ↓. A larger sample of stations measuring incident radiation with a high quality is needed evaluate how well the models capture trends within the region, particularly given the uncertainty in the observational data.

## 8 Conclusions

Hydrological, ecological, and crop modellers seek landscape scale data. Norway has a long coastline with mountains, fjords and small islands. Strong land-sea contrast, high mountains, and a seasonal snow cover highly dependent on continentality and altitude results in a fine scale variability difficult for coarse scale models to represent. A Python script to downscale and consolidate reanalysis data with high resolution national gridded temperature data has been developed; which, leaning on



previously well tested empirical relationships, provides estimates of humidity and incident radiation on a fine scale grid. The downscaled humidity ensures that relative humidity is constrained at 100 % so that e.g. reasonable evaporation estimates can be sought. The new estimates, HySN, provide humidity estimates with the overall highest quality given the metrics considered here; surpassing also those based on estimating humidity from temperature alone, such as for the VIC Type Forcing data. The

new estimates outperform the VIC Type Forcing data and the MERRA-2 estimates of incident radiation, however, it is not clear that the new estimates have an added value compared to Era-Interim and WFDEI. The lack of high quality historical observations, particularly of incident longwave radiation hinders a proper evaluation of the data sets.

-Additionally, this study has shown that ($\mathcal{H}_a$) altitude is a significant predictor of humidity, SW ↓ and LW ↓ in the domain. The coarse scale estimates of $T_d$ increased on average with 1 °C, SW ↓ with 0.7-1.5 $\mathrm{Wm}^{-2}$, and LW ↓ was increased with as

much as 8.6 $\mathrm{Wm}^{-2}$, when adjusted to station altitude.

-Further, the results have shown that a high resolution does not necessarily indicate high quality estimates. The added value of the high horizontal resolution of the more empirically based estimates does not outweighs the added value of relying on estimates from coarser resolution numerical weather prediction reanalyses ($\mathcal{H}_b$). Not only is a higher daily temporal correlation (ACC) seen in the estimates based on newer reanalysis data compared to the VIC Type Forcing data, but also a lower mean

absolute station bias is seen for several reanalysis based products (Era, WFDEI, HySN). VFDv1 and VFDv2 shows a 60 % stronger decrease of humidity with distance from the coast than the observations, alluding to that the modified version of the Magnus-type formula based on Kimball et al. (1997), implemented in VFD to estimate humidity from daily minimum temperature, is not appropriate for the Norwegian domain. Both VFDv1 and VFDv2 also show a several times stronger decrease in solar radiation with latitude than the observations, likely a result of using diurnal temperature range as a proxy for cloud

cover, an assumption likely not appropriate in coastal environments and at high latitudes.

To our knowledge reanalysis based estimates have not been compared with VIC Type Forcing Data for regions within Europe, or Norway specifically. The comparison of model estimates may assist impact modellers which have not yet selected data to use. For some the findings might help explain persistent errors, for instance found in the timing of snowmelt in a hydrological model. The findings provides emphasis for climate researchers to not only downscale $T_2$ and precipitation from

climate projections and later use these to estimate humidity and incident radiation, but to utilize the model estimates of near surface humidity and incident radiation, as is already done e.g. in ISI-MIP.

The source code for computing HySN has been made available may easily be configured to use other reanalysis data or other national data sets as input. The compilation of 36 years of daily estimate of humidity, surface pressure, LW ↓, and SW ↓ requires merely half a day on a modern desktop computer. Part of the code might also be implemented in a model pre-processor

or in the calculation of various index, so that the variables do not need to be stored for long time spans. Future work entail calculating indices such as reference evaporation, and updating the input data to Era5 and a new version of SeNorge once the full historical time-series of the two are available. Additionally, sub-daily estimates, the inclusion of terrain features such as slope and aspect, and adding a correction based on the lack of coupling between the land surface and the atmosphere at times when the differences in the local snow cover and that modelled by the reanalysis is large might be promising, as initial results



showed that differences in ground snow conditions between the reanalysis and the observations were significant in predicting the difference between Era-Interims estimates of SW ↓ and the observations.

# 9  Code and data availability

The HySN data product is available archived in Zenodo (https://doi.org/10.5281/zenodo.1970170). The code used to produce the HySN estimates is written in Python and is available at https://github.com/helene-b-e/HySN.git; further the particular version of the software code used to produced the HySN estimates validated here is archived in Zenodo (https://doi.org/10.5281/zenodo.1435555). The remaining data sets are available from the various data providers.

*Author contributions.* All authors discussed the results and contributed to the manuscript. HBE derived the HySN dataset, acquired and quality controlled data from external sources, analysed the results, and drafted the manuscript.

*Competing interests.* No competing interests are present.

*Acknowledgements.* We would like to thank Sigbjørn Grini for providing the scripts for quality control of the SW ↓. We further thank all data providers: MET Norway, NIBIO, UiB (where radiation measurements were provided by Jan Asle Olseth), NASA, and ECMWF, and thank Ingjerd Haddeland, Jan Magnusson, and Shaochun Huang for compiling the VIC Forcing Data while working at NVE. The PGMFDv2 was downloaded from the ISIMIP node of the ESGF server (https://esg.pik-potsdam.de/projects/isimip/). The study forms a contribution to LATICE, which is a strategic research area founded by the Faculty of Mathematics and Natural Sciences at the University of Oslo. Helene B. Erlandsen was founded by NVE.





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





**Table 1.** The following data sets provide estimates of humidity, LW ↓, and SW ↓ which are evaluated. Precipitation is denoted as P, 2-meter temperature as $T_2$. The global data sets are retrieved from online repositories, while the data sets with regional coverage are compiled locally, based on the stated input data; the VFD-datesets using the VIC model's pre-processor, and the HySN-data set based on the methods outlined in the current study. Additional references are given elsewhere in the text.

| | Product | Resolution | Coverage | Type | Processing methods | Input data | Surface obs. |
|---|---|---|---|---|---|---|---|
| M | MERRA2 | 1/2°x 2/3° | Global | Reanalysis | | | No |
| E | Era-Interim | 2/3°x 2/3° | Global | Reanalysis | | | Yes |
| P | PGMFDv2 | 1/2°x 1/2° | Global, land only | Post-processed reanalysis | -VP, LW ↓ re-gridded and adjusted to monthly CRU $T_2$, method from Cosgrove (2003) - SW ↓ & LW ↓ adjusted to satellite-based data set | -NCEP-NCAR (2°x 2°), -CRU TS3.1 $T_2$, cloud cover -NASA MEaSUREs LW ↓ & SW ↓ | Yes |
| W | WFDEI | 1/2°x 1/2° | Global, land only | Post-processed reanalyis | -VP, LW ↓ re-gridded and adjusted to monthly CRU $T_2$, method from Cosgrove (2003) -SW ↓ re-gridded and adjusted to CRU cloud cover & inter-annual aerosol loading | -ERA-Interim -1979-2009: CRU TS3.1 $T_2$ -CRU cloud cover and aerosol loading | Yes |
| H | HySN | 1 x 1 km | Regional, locally compiled | Post-processed reanalyis | -VP, LW ↓ re-gridding and adjustment to daily SeNorge2 $T_2$, method from Cosgrove (2003) -SW ↓ re-gridding and adjustment, method from Thornton and Running (1999) | -ERA-Interim -SeNorge2 $T_2$ | Yes |
| V1 | VFDv1 | 1 x 1 km | Regional, locally compiled | Empirical model | running the VIC4.0.6 pre processor: MTCLIMv4.2, TVA+Bras LW ↓ | -SeNorge P, $T_2$ -Nora10 sub-daily $T_2$ | Yes |
| V2 | VFDv2 | 1 x 1 km | Regional, locally compiled | Empirical model | running the VIC4.2.d pre -processor: MTCLIMv4.3, Prata+Deadroff LW ↓ | -SeNorge2 P, $T_{2min}$, $T_{2max}$ | Yes |



**Table 2.** An overview of the automatic quality control tests, based on the relative values of the solar zenith angle (sza), measured (SW), extraterrestrial ($SW_E$), and clear sky ($SW_{CS}$) incident global radiation. The table is adapted from Table 4.1.1 in Grini (2015).

| Name | Quality requirement | Quality procedure |
|---|---|---|
| Offset | $SW \leq$-12 $Wm^{-2}$ | Visual control |
| | $SW <$6 $Wm^{-2}$ if sza $< 93°$ | of flagged data |
| Upper bound 1 | $SW < SW_E$ | Flagged as erroneous |
| Upper bound 2 | $SW \leq 1.1\ SW_{CS}$ if sza $< 88°$ $SW \leq 2\ SW_{CS}$ if sza $\geq 88°$ | Flagged as erroneous |
| Lower bound 1 | $\mu \frac{SW}{SW_E} \leq 0.03$ | The day flagged as erroneous |
| Lower bound 2 | $SW \leq 10^{-4}(80\text{-sza})\ SW_E$ if sza $\leq 80°$ | Flagged as erroneous |

**Table 3.** Summary of metrics showing the humidity estimates similarity to station observations. Differences ($\Delta$) are given in dew point temperature in $°C$. $\Delta$ is the mean station difference, $|\Delta|$ is the mean absolute station difference, $|\delta|_{max}$ is the largest absolute difference at any station, while $|\delta^s|_{max}$ is the largest seasonal difference at any station. ACC is the anomaly (de-seasonalized) daily correlation coefficient, while K-S indicates the number of station where the daily mean cumulative distribution passes the Kolmogrov-Smirnov test of similarity (p>0.001).

| Model | $\Delta$ | $|\Delta|$ | $|\delta|_{max}$ | $|\delta^s|_{max}$ | ACC | K-S |
|---|---|---|---|---|---|---|
| MERRA2 | 1.4 | 1.5 | 4.1 | 4.7 | 0.79 | 0 % |
| Era-Interim | 0.9 | 1.0 | 3.7 | 4.4 | **0.86** | 10 % |
| PGMFDv2 | 1.7 | 1.8 | 5.4 | 6.2 | 0.52 | 0 % |
| WFDEI | 0.7 | 0.9 | 3.3 | 3.9 | 0.85 | **15 %** |
| VFDv1 | -0.7 | 1.0 | -4.2 | -6.1 | 0.58 | 5 % |
| VFDv2 | -1.0 | 1.2 | -5.3 | -7.2 | 0.66 | 3 % |
| HySN | **0.1** | **0.7** | **2.8** | **3.7** | 0.83 | **15 %** |

**Table 4.** As in Table 3, but with metrics listed for SW ↓. Except for ACC and K-S, which are dimensionless, the units are $Wm^{-2}$.

| Model | $\Delta$ | $|\Delta|$ | $|\delta|_{max}$ | $|\delta^s|_{max}$ | ACC | K-S |
|---|---|---|---|---|---|---|
| MERRA2 | 13 | 13 | 32 | 49 | 0.73 | 10 % |
| Era-Interim | 4 | **4** | 19 | 20 | **0.78** | 60 % |
| PGMFDv2 | 11 | 11 | 19 | 38 | 0.31 | 0 % |
| WFDEI | **2** | **4** | **8** | **19** | 0.76 | 60 % |
| VFDv1 | -4 | 10 | -23 | -62 | 0.48 | 10 % |
| VFDv2 | 9 | 10 | 26 | 54 | 0.40 | 20 % |
| HySN | 3 | **4** | 9 | 20 | **0.78** | **70 %** |



**Table 5.** As in Table 3, but with metrics listed for LW ↓. Except for ACC and KS, which are dimensionless, the units are $\mathrm{Wm}^{-2}$.

| Model | $\Delta$ | $|\Delta|$ | $|\delta|_{\max}$ | $|\delta^s|_{\max}$ | ACC | K-S |
|---|---|---|---|---|---|---|
| MERRA2 | -14 | 14 | -20 | -24 | 0.81 | 0/2 |
| Era-Interim | -8 | 8 | -13 | -14 | 0.79 | **1/2** |
| PGMFDv2 | -10 | 10 | -12 | -16 | 0.46 | 0/2 |
| WFDEI | **6** | **6** | **11** | **12** | 0.82 | **1/2** |
| VFDv1 | -9 | 9 | -11 | -15 | 0.51 | 0/2 |
| VFDv2 | -15 | 15 | -17 | -21 | 0.60 | 0/2 |
| HySN | -11 | 11 | -15 | -17 | **0.84** | 0/2 |

**Table 6.** Linear, decadal trends in monthly mean $\mathrm{T_d}[°C]$ between January 1985 and December 1999, significant at a 5 % or 10 % (denoted in *italics*) level in the observations (O) and the model estimates, listed with the month and region denoted on the left (month, region).

| | O | M | E | P | W | V | V2 | H |
|---|---|---|---|---|---|---|---|---|
| $\mathrm{Apr, S_W}$ | *1.7* | | | | | | | |
| $\mathrm{Apr, S_E}$ | 2.2 | | | | | | | |
| $\mathrm{May, S_W}$ | | *-1.2* | | *-1.5* | | | | *-1.1* |
| $\mathrm{May, S_E}$ | | | | -1.8 | | | | |
| $\mathrm{May, C}$ | *-1.3* | -1.2 | -1.4 | -1.3 | *-1.3* | | | -1.5 |
| $\mathrm{May, N_E}$ | | *-0.9* | | | | | | |
| $\mathrm{Jul, N_E}$ | 1.6 | | | | | | | |
| $\mathrm{Aug, S_W}$ | | *1.1* | | | | *1.0* | | |
| $\mathrm{Aug, S_E}$ | | | | | | *1.6* | | |
| $\mathrm{Sep, S_W}$ | 2.7 | 2.0 | 2.4 | 2.3 | 2.3 | 2.6 | 2.4 | *2.2* |
| $\mathrm{Sep, S_E}$ | 3.0 | 2.6 | 3.0 | 2.3 | 3.0 | 3.2 | 2.9 | 3.3 |
| $\mathrm{Sep, C}$ | 2.1 | 1.9 | 1.9 | | 1.8 | 2.3 | *1.9* | 2.0 |
| $\mathrm{Sep, N_W}$ | *1.8* | *1.6* | | | | *1.6* | | |
| $\mathrm{Oct, C}$ | | | | *-1.6* | | | | |

**Table 7.** Linear, decadal changes in monthly mean $\mathrm{T_d}$ [$°C$] for Bergen-Florida station, and for the co-located grid cells of the model estimates between January 1985 and December 1999, significant at a 5 % or 10 % (*italics*) level.

| [h!] | O | M | E | P | W | V1 | V2 | H |
|---|---|---|---|---|---|---|---|---|
| Apr | 2.0 | | | | | *1.2* | *0.9* | |
| May | | *-1.3* | -1.4 | | | | | -1.4 |
| Jul | 1.7 | | | | | | | |
| Aug | 1.9 | | | | | | | |
| Sep | 3.2 | 2.3 | | 2.3 | 2.1 | 3.1 | 2.6 | 2.2 |



**Table 8.** Linear, decadal changes in monthly mean SW ↓ [Wm$^{-2}$] at Bergen between January 1985 and December 1999, significant at a 5 % or 10 % (denoted in *italics*) level.

|     | O  | M  | E  | P    | W | V1 | V2   | H  |
|-----|----|----|----|------|---|----|------|----|
| Jan |    |    |    | 1    |   |    |      |    |
| Aug | 51 | 26 | 47 | 42   |   |    | *25* | 46 |
| Sep |    |    | 24 | *15* |   |    |      | 23 |
| Oct | 7  |    |    | 7    |   |    |      |    |
| Nov |    |    | 3  |      |   |    |      |    |
| Dec | 1  | *1* | 2 |      |   | *1* |     |    |

**Table 9.** Linear, decadal changes in monthly mean LW ↓ [Wm$^{-2}$] at Bergen between January 1985 and December 1999 significant at a 5 % or 10 % (denoted in *italics*) level.

|     | O    | M | E | P    | W   | V    | V2 | H    |
|-----|------|---|---|------|-----|------|----|------|
| Apr |      |   |   |      |     | 8    |    |      |
| May | -21  |   |   | *-12* |    |      |    |      |
| Aug | -13  |   |   |      | -12 |      |    | *-11* |
| Sep |      |   |   |      |     | *14* | 14 |      |
| Oct | *-17* |  |   |      |     |      |    |      |
| Dec | -17  |   |   |      |     |      |    |      |