# Peer review of "Merits of novel high-resolution estimates and existing long-term estimates of humidity and incident radiation in a complex domain"

_Earth System Science Data, 2018_

## Referee Comment (RC1) · Robinson (Referee) · 28 Jan 2019

This paper summarizes a new dataset, HySN, of three variables (humidity, short- and long-wave radiation), which are very useful for a variety of modelling applications, but are historically not as well defined as eg temperature and precipitation. The paper provides a description of the production of the data, and carries out a thorough evaluation against observations and other modelled products of the same variables.

In general, the paper is well-structured and clearly written. It demonstrates the benefit

of the new dataset compared to existing modelled gridded products, especially due to the increase in resolution. My overall impression is very positive but I have some minor concerns/suggestions.

**1 Minor corrections:**

**Abstract:** Somewhere you should mention the full temporal extent of the dataset. From the paper it sounds like you have only produced data for 1982-2000, but looking at the Zenodo link, the data are available from 1979-2017.

**P1 L13:** Td has not yet been defined, either define it here or just write dew-point temperature.

**P2 L18-19:** "In recent time a gridded, observation-based data set of near surface wind speed has also been developed." Please provide a reference for this.

**P6 L23:** "The data is currently compiled for the time period 1982-2000..." This is confusing, as the Zenodo archive seems to contain data for 1979-2017. If you are only choosing to analyse the years 1982-2000 for this paper, please explain why (due to availability of observations?). But don't do yourself a disservice by not advertising the whole dataset!

**P8 Fig 1:** The description of the markers is a little confusing – it sounds like there are multiple purple markers rather than that the purple marker is a site for which both SW$_\downarrow$ and LW$_\downarrow$ are available. Could you rephrase this? Also, the blue-red colour bar for the difference plot has the numbers overlapping the units, please adjust this.

**P9 L9-10:** This sounds like you are referring to Table 2 in Grini (2015), you should make it clear that this is referring to Table 2 in this paper.

**P12 L17:** Why does the trend analysis only start in 1985, when the rest of the analysis

uses 1982 onwards?

**P13 L13:** Why is this not shown? Maybe add something to the supplementary information to support this statement.

**P15 Figure 7:** In the caption you refer to "upper", "middle" and "lower" plots. Please change this to left, middle and right as they are arranged.

**P21 L10:** Why are you now only using the data up to 1999? Please explain.

**P21 L26:** You refer to Table 8 before Table 7, please check the numbering. Also, you never refer to Table 9 in the text, please do so.

**P28 L28:** Why 36 years?

**2  Typographic errors:**

**Throughout:** "Era-Interim" should be capitalised: "ERA-Interim".

**Throughout:** You sometimes write "T2" and sometimes "$T_2$", please be consistent through the manuscript. Similarly, please choose between "Td" and "$T_d$".

**P3 L29:** Should read "based on reanalysis data".

**P6 L6:** Lussana et al should be within the parentheses.

**P10 L5:** I think "Table S.2" should be "Table A3".

**P10 L20:** "staring point" should be "starting point".

**P11 L12:** You are losing some of the equation off the side of the page.

**P11 L14:** "$\mu_{station,station,observed}$" should be "$\mu_{season,station,observed}$"

**P12 L20:** "$SW_\downarrow$" should just be "SW" in this case (south-west, not short-wave).

**P28 L8-14:** Please format lists properly (indented, with bullets).

**P28 L30:** "entail" should be "entails".

**P36 Table 1:** The hyphens/minus signs at the start of each sentence in the "Processing methods" and "input data" columbs are quite confusing. If these are supposed to be points in lists, please format the lists appropriately with bullet points.

**P37-39:** The tables are not consistently formatted, they should really have horizontal lines at top, bottom and between header and content. Also, what does it mean that some numbers are bold in Tables 3-5?

---

## Referee Comment (RC2) · Weedon (Referee) · 31 Jan 2019

The authors have set out to determine if there are vertical gradients in near surface humidity and down-welling radiation in Norway (Ha) and to test whether there is added value to generating high-resolution (0.1 x 0.1 degree) estimates of humidity, SWdown and LWdown for this country (Hb). They documented their processing of meteorological observations from across Norway, generated new data (HySN) and then compared their results to observations, raw surface reanalyses data (from MERRA and ERA-Interim), bias-corrected reanalyses (PGF and WFDEI) and two VIC-generated forcing

datasets.

This is a careful and well written piece of work. Although they clearly showed that humidity and down-welling radiation have significant vertical gradients (Ha), they also conclude that to an extent there is not a huge added value obtained by generating the new data (Hb). These are justified conclusions, but they could choose to amplify their results a little as explained below.

They have justified their effort by indicating that the high-resolution forcing data will be of value for land surface models including Norwegian operational hydrological models. They have shown that new data, assessed as isolated meteorological variables, out-perform existing datasets in many situations. However, although this is of course primarily a paper about a dataset, of additional value to the reader would be at least a partial test of whether the differences between the various datasets cause large differences to the output from a hydrological model.

Therefore I would urge the authors to select a single catchment for which they have observations of discharge and then to simply illustrate whether there is much spread in the discharge estimates from a single hydrological model according to the various forcing datasets for SWdown, LWdown and humidity - but with every model run using the same 2m temperature and precipitation forcing. Hopefully this would not be too onerous (one model, one catchment, one observational dataset, seven model runs) and would clearly illustrate the relative significance of a spread in forcing in terms of SWdown, LWdown and humidity to the performance of a hydrological model.

It is entirely possible that the authors already plan to publish this type of model analysis so I do not wish to imply that I think it is essential for this paper, but I do think it would provide more impact. It could be a starting point for further exploration in a subsequent paper (multiple catchments, isolating impact of LWdown, SWdown and humidity separately and possibly multiple hydrological models).

The minor corrections below mainly relate to typographic errors plus ways in which

Figs 1f, 1g, 3 and 7 could be made much easier to digest.

Minor corrections: p1 line 22: a latent heat flux, which in turn > a sensible heat flux as well as a latent flux which in turn

p4 line 7: restrains currently > restraint, currently

p4 line 29: MT-CLIM > MTCLIM

p6 line 10: WATCH > WFDEI [Note: WFDEI was entirely funded by EU-EMBRACE. EU-WATCH had finished by the time it was created].

p7 lines 7-8: Bras full-sky LW v algorithm. Provide a reference describing this algorithm.

p7 line 18: snow cover the > snow cover in the

p7 line 7: snow season varying from a few days to 300 days a year. Provide an explanation in brackets for this large range for those who don't know Norway. E.g.: "(dependent on latitude, elevation and distance from the coast)".

p8 line 4: The quality of the observations are fair . . . "Fair" is far too vague to be helpful – it might mean the results are reliable to within +/-5 or +/- 50%! Please provide a quantitative estimate of reliability.

p8 line 7: red and purple. There is no good reason to use two colours for LW V sites in Fig 1g. See the notes on Figure changes below – I suggest using one colour that can easily be seen such as orange.

p9 line 9: conducted this study > conducted for this study

p9 line 14: time-series > time series [two words (e.g. see primary mathematical literature) only hyphenate when writing time-series analysis]

p10 line 9: located at the ground > located on the ground

p10 line 13: time-series > time series

p10 line 18: to not introduce spatial or temporal smoothing > to avoid introducing spatial or temporal smoothing

p10 line 23: analysed multiple > analysed using multiple

p11 lines 11 to 14: Use separate lines for each equation. Indicate that "mu" = mean.

p11 line 15: time-series > time series

p11 line 18: probability of the that the > probability that the

p11 line 24: differs significantly > differ significantly [dependencies and model esti-mates are both plural in this sentence]

p12 line 2: a air mass > an air mass

p12 line 16: time-series > time series

P13 line 19: weakened with 0.11 hPa/100 m > weakened by 0.11 hPa/100 m

p16 line 21: fall is > fall (autumn) is

p20 line 21: trough > through

p22 line 22: higher found in > higher than found in

p23 line 11: average to a five times larger > average in a five times larger

p25 line 12: produced points to that great care > produced indicates that great care

P25 line 23: sensitivity too > sensitivity to

p27 line 9: conducting in the production > conducted in the production

p27 line 12: which might to do > which might be to do

p27 line 16: does not implies > does not imply

p28 line 26: e.g. in ISI-MIP. Provide reference to ISI-MIP research.

p28 line 27: made available may easily > made available and may easily

Figure changes: Fig. 1f and 1g The resolution of the background map plus the background colours (green, yellow, white) make it very hard to see the plotted points in blue and purple. I suggest removing the background colour in 1f and plot points in red. There is no need to plot the LW V stations in different colours in 1g. Use a colour that stands out instead (e.g. orange for LW V sites).

Fig 2 Make sure the observations stand out. I suggest using a thick continuous black line instead of a thin dashed line lacking plotted points. Increase the size of all the text (including key, heading and axis label) relative to the figure. Add a horizontal axis label "Calendar month".

Fig 3. It is extremely difficult to distinguish these lines. Avoid using red and green on the same plot. Remove all the plotted shapes – it confuses things. I suggest instead using a thick continuous line to make the observations easily distinguished. For the models use fewer colours according to dataset type (e.g. MERRA and ERA-Interim = red; PGF and WFDEI = blue; V1 and V2 = orange, HySN = grey). Then distinguish within pairs of lines of the same dataset type by using one continuous and one dashed (e.g. MERRA continuous red, ERA-Interim dashed red). Add explanation of the key letters to the caption.

Fig. 5 Make plotted symbols much bigger.

Fig 6. Use system suggested for fig. 3.

graham.weedon@metoffice.gov.uk 31st Jan 2019
* * *

---

## Author Comment (AC1) · 27 Feb 2019

The authors would like to thank the reviewer, Emma Robinson, for the thorough and positive review, with detailed suggestions, which, once implemented, will result in a clearer and improved manuscript. The suggested corrections will be implemented and the typographical errors corrected in the revised manuscript.

---

## Author Comment (AC2) · 27 Feb 2019

Thank you for the thorough and positive review as well as constructive comments regarding the work and its potential improvements. The implementation of your suggested minor corrections, particularly those concerning the figures, are of great use and will improve the readability of the revised manuscript.

The suggestion of including a (partial) test of the various forcing data-sets presented in a hydrological model is well considered; and is something we authors have debated in

the earlier stages of the project. We did, however, arrive at the conclusion to introduce the constructed data-set in a separate data-set journal, and to apply the data in another study. This is because a single application might derail the reader, since there are many possible applications of the data-set, for instance that of acting as historical reference data. The other aspect we have considered is that - if the application should have a significant value - the design of the experiment and interpretation of its result require a thorough treatment and thus, we would argue that it should be presented as a separate study.

Applying the different data-sets in a hydrological model would entail choosing (1) whether or not to run the hydrological model at the scale of the forcing data, (2) how to up-scale the SeNorge data or how to down-scale the coarse scale humidity and radiation data in Norway's complex terrain, (3) whether scale itself should be considered when analysing the results (particularly given the influence of snow in Norway, which has a non-linear sensitivity to temperature/the surface energy balance), and (4) if the model should be calibrated separately for each of the different data-sets of humidity and radiation and possibly for the different scales it would be run at. We will highlight the testing of the various forcing data-sets in a hydrological model as a valuable future study in the revised version of the manuscript.

---

## Author Response (AR1)

The authors would again like to thank the reviewer, Emma Robinson, for taking the time to review this work, and for a thorough and positive review. Our response to the reviewer comments are marked with "Response", further the reviewers comments are marked with RC1 and with italics.  A marked-up manuscript version showing the changes made is appended after the response.

*RC1: Interactive comment on "Merits of novel high-resolution estimates and existing long-term estimates of humidity and incident radiation in a complex domain" by Helene Birkelund Erlandsen et al.*
*Robinson (Referee) emrobi@ceh.ac.uk*

*RC1: This paper summarizes a new dataset, HySN, of three variables (humidity, short- and long-wave radiation), which are very useful for a variety of modelling applications, but are historically not as well defined as eg temperature and precipitation. The paper provides a description of the production of the data, and carries out a thorough evaluation against observations and other modelled products of the same variables. In general, the paper is well-structured and clearly written. It demonstrates the benefit C1 ESSDD Interactive comment Printer-friendly version Discussion paper of the new dataset compared to existing modelled gridded products, especially due to the increase in resolution. My overall impression is very positive but I have some minor concerns/suggestions.*

***RC1: Minor corrections****:*

*RC1: Abstract: Somewhere you should mention the full temporal extent of the dataset. From the paper it sounds like you have only produced data for 1982-2000, but looking at the Zenodo link, the data are available from 1979-2017.*
Response: Thank you for noticing this; the full range of years which HySN covers is now mentioned in the abstract.

*RC1: P1 L13: Td has not yet been defined, either define it here or just write dew-point temperature.*
Response: This has been updated to state dew-point temperature.

*RC1: P2 L18-19: "In recent time a gridded, observation-based data set of near surface wind speed has also been developed." Please provide a reference for this.*
Response: There is not yet a proper reference for this data set. The data set was constructed by researchers at the Norwegian Meteorological Institute, primarily Dr. Thomas Nipen. The high

resolution wind data set was constructed by bias-correcting wind speeds from Nora10 (Reistad, 2011) using a higher resolution reference dataset. In particular, a quantile mapping approach using the AROME MetCoOp weather model (Müller et al, 2017) at 2.5 km grid spacing as reference. The quantile maps were based on Nora10 analyses and AROME forecasts from 2013-2015 and were computed separately for each gridpoint. The data are available from the the Norwegian Meteorological Institute's thredds-server under the folder FFMRR-Nor/ (http://thredds.met.no/thredds/catalog/metusers/klinogrid/KliNoGrid_16.12/catalog.html). The updated manuscript includes a brief description of the approach, including the weblink.

*RC1: P6 L23: "The data is currently compiled for the time period 1982-2000. . ." This is confusing, as the Zenodo archive seems to contain data for 1979-2017. If you are only choosing to analyse the years 1982-2000 for this paper, please explain why (due to availability of observations?). But don't do yourself a disservice by not advertising the whole dataset!*
Response: The line has been updated to 1979 - 2017. The comparison to observations in the period 1982 - 2000 was done due to the availability of the MTCLIM data sets.

*RC1: P8 Fig 1: The description of the markers is a little confusing – it sounds like there are multiple purple markers rather than that the purple marker is a site for which both SW↓ and LW↓ are available. Could you rephrase this? Also, the blue-red colour bar for the difference plot has the numbers overlapping the units, please adjust this.*
Response: The plot has been updated according to comments from both you and the second reviewer. Both LW markers are now in orange, while the text mentions that the southernmost LW station also measures SW.

*RC1: P9 L9-10: This sounds like you are referring to Table 2 in Grini (2015), you should make it clear that this is referring to Table 2 in this paper.*
Response: This now reads "For stations other than Bergen, quality control procedures , follow the methodology suggested by Grini (2015) as outlined in Table 2."

*RC1: P12 L17: Why does the trend analysis only start in 1985, when the rest of the analysis uses 1982 onwards?*
Response: The starting year, 1985, was chosen as this often is considered the start of the surface solar brightening period in Europe following a dimming period likely to a large degree, due to aerosol emissions. This was mentioned in the Results section of the manuscript (P21 L9), but is now also included in Section 5, Evaluation methods, in the revised version of the manuscript.

*RC1: P13 L13: Why is this not shown? Maybe add something to the supplementary information to support this statement.*
Response: The supplementary information is now updated with an additional section, Sec. 2, containing  a brief paragraph and three figures showing the slopes between vapour pressure and altitude and their significances, for annual mean values at or near (the nearest neighbour

grid cell of) the observation stations. The figures show the vertical gradients for three different regression models, where the number of geographical predictors are varied.

*RC1: P15 Figure 7: In the caption you refer to "upper", "middle" and "lower" plots. Please change this to left, middle and right as they are arranged.*
Response: Corrected.

*RC1: P21 L10: Why are you now only using the data up to 1999? Please explain.*
Response: Thank you for noticing this. The time range considered in the analysis and comparison to observations is from the 1st of January 1982 until the 31st of December 1999, and was denoted as 1982-2000. This has been updated to rather state from 1982 through 1999 throughout, with the trend analysis covering monthly means between January 1985 until December 1999, as stated.

*RC1: P21 L26: You refer to Table 8 before Table 7, please check the numbering. Also, you never refer to Table 9 in the text, please do so.*
Response: This has been updated.

*RC1: P28 L28: Why 36 years?*
Response: This has been updated to plainly state "The compilation of HySN requires merely half a day …"

***RC1: 2 Typographic errors:***

*RC1: Throughout: "Era-Interim" should be capitalised: "ERA-Interim".* Response: Corrected.

*RC1: Throughout: You sometimes write "T2" and sometimes "T_2", please be consistent through the manuscript.* Response: Updated to T_2

*RC1: Similarly, please choose between "Td" and "Td".* Response: Ok.

*RC1: P3 L29: Should read "based on reanalysis data".* Response: Corrected.

*RC1: P6 L6: Lussana et al should be within the parentheses.* Response: Corrected.

*RC1: P10 L5: I think "Table S.2" should be "Table A3".* Response: Corrected.

*RC1: P10 L20: "staring point" should be "starting point".* Response: Thanks. This has been corrected.

*RC1: P11 L12: You are losing some of the equation off the side of the page.* Response: The equations are now included in a list of the metrics. Hopefully this resolves the formatting issue.

*RC1: P11 L14: "µstation,station,observed" should be "µseason,station,observed"* Response: Corrected.

*RC1: P12 L20: "SW↓" should just be "SW" in this case (south-west, not short-wave).* Response: The regions are now abbreviated as they were in Table 6 of the manuscript, with West and East denoted in subscript.

*RC1: P28 L8-14: Please format lists properly (indented, with bullets).* Response: Updated

*RC1: P28 L30: "entail" should be "entails".* Response: Corrected.

*RC1: P36 Table 1: The hyphens/minus signs at the start of each sentence in the "Processing methods" and "input data" columbs are quite confusing. If these are supposed to be points in lists, please format the lists appropriately with bullet points.* Response: The hyphens have been removed.

*RC1: P37-39: The tables are not consistently formatted, they should really have horizontal lines at top, bottom and between header and content. Also, what does it mean that some numbers are bold in Tables 3-5?* Response: Thank you for noticing this. The horizontal lines have been added where they were missing. The best scores are shown in bold. This is now stated in the caption.

References added

Müller, M., M. Homleid, K. Ivarsson, M.A. Køltzow, M. Lindskog, K.H. Midtbø, U. Andrae, T. Aspelien, L. Berggren, D. Bjørge, P. Dahlgren, J. Kristiansen, R. Randriamampianina, M. Ridal, and O. Vignes, 2017: AROME-MetCoOp: A Nordic Convective-Scale Operational Weather Prediction Model. *Wea. Forecasting,* **32**, 609–627, https://doi.org/10.1175/WAF-D-16-0099.1

Reistad, M., Breivik, Ø., Haakenstad, H., Aarnes, O. J., Furevik, B. R., and Bidlot, J.-R. ( 2011), A high-resolution hindcast of wind and waves for the North Sea, the Norwegian Sea, and the Barents Sea, *J. Geophys. Res.*, 116, C05019, doi:10.1029/2010JC006402

Authors response to *Interactive comment on "Merits of novel high-resolution estimates and existing long-term estimates of humidity and incident radiation in a complex domain" by Helene Birkelund Erlandsen et al. by Weedon (Referee)*

The authors would like to thank you for taking the time to review this work, and for the overall positive comments with helpful and constructive suggestions. We have addressed your comments one by one in the following. Our responses are marked with "Response", and the reviewers comments are marked with RC2 and italicized. A marked-up manuscript version showing the changes made is appended after the response.

*RC2: The authors have set out to determine if there are vertical gradients in near surface humidity and downwelling radiation in Norway (Ha) and to test whether there is added value to generating high-resolution (0.1 x 0.1 degree) estimates of humidity, SWdown and LWdown for this country (Hb). They documented their processing of meteorological observations from across Norway, generated new data (HySN) and then compared their results to observations, raw surface reanalyses data (from MERRA and ERAInterim), bias-corrected reanalyses (PGF and WFDEI) and two VIC-generated forcing datasets. This is a careful and well written piece of work.*

*Although they clearly showed that humidity and down-welling radiation have significant vertical gradients (Ha), they also conclude that to an extent there is not a huge added value obtained by generating the new data (Hb). These are justified conclusions, but they could choose to amplify their results a little as explained below. They have justified their effort by indicating that the high-resolution forcing data will be of value for land surface models including Norwegian operational hydrological models. They have shown that new data, assessed as isolated meteorological variables, out-perform existing datasets in many situations. However, although this is of course primarily a paper about a dataset, of additional value to the reader would be at least a partial test of whether the differences between the various datasets cause large differences to the output from a hydrological model. Therefore I would urge the authors to select a single catchment for which they have observations of discharge and then to simply illustrate whether there is much spread in the discharge estimates from a single hydrological model according to the various forcing datasets for SWdown, LWdown and humidity - but with every model run using the same 2m temperature and precipitation forcing. Hopefully this would not be too onerous (one model, one catchment, one observational dataset, seven model runs) and would clearly illustrate the relative significance of a spread in forcing in terms of SWdown, LWdown and humidity to the performance of a hydrological model. It is entirely possible that the authors already plan to publish this type of model analysis so I do not wish to imply that I think it is essential for this paper, but I do think it would provide more impact. It could be a starting point for further exploration in a subsequent paper (multiple catchments, isolating impact of LWdown, SWdown and humidity separately and possibly multiple hydrological models).*

Response: This is a good suggestion and is along the lines of an application we discussed including at an earlier stage of the study. However, after going through all the data processing

stages and choices needed to design an interesting application of the data, we realized that the amount of decisions leading up to such a study deserved a proper description, and more space than we felt it suitable to provide in the current paper and journal. We are currently working on a separate study applying the new dataset in a conceptual hydrological model, which also includes altering the hydrological model's code to become more physical based in the description of several processes. It is not given that the inclusion of new variables from HySN in a conceptual hydrological model with a considerable amount of tunable parameters calibrated against runoff, will have a substantial impact on runoff metrics. We believe that a greater value of providing these additional reference data at a resolution relevant for landscape modelling, is the opportunity for the landscape-scale modelers to modulate their code to include a higher degree of realism, i.e. a sensitivity to other variables than e.g. precipitation and temperature where relevant, so that the models may more robustly react to e.g. land use change or climate change. The new study will likely also include a sensitivity test of the hydrological models to variations in forcing data. In the revised manuscript, P29 L11-14 we have included a description of such a study as planned future work.

*RC2: The minor corrections below mainly relate to typographic errors plus ways in which Figs 1f, 1g, 3 and 7 could be made much easier to digest.*

***RC2: Minor corrections:***

*RC2: p1 line 22: a latent heat flux, which in turn > a sensible heat flux as well as a latent flux which in turn* Response: This has been updated.

*RC2: p4 line 7: restrains currently > restraint, currently* Response: Corrected.

*RC2: p4 line 29: MT-CLIM > MTCLIM*
Response: Updated throughout

*RC2: p6 line 10: WATCH > WFDEI [Note: WFDEI was entirely funded by EU-EMBRACE. EU-WATCH had finished by the time it was created].*
Response: Updated.

*RC2: p7 lines 7-8: Bras full-sky LW v algorithm. Provide a reference describing this algorithm.*
Response: A reference to the textbook which contains *the Bras full-sky LW algorithm i*s included in the revised manuscript.

*RC2: p7 line 18: snow cover the > snow cover in the*
Response: Updated

*RC2: p7 line 7: snow season varying from a few days to 300 days a year. Provide an explanation in brackets for this large range for those who don't know Norway. E.g.: "(dependent on latitude, elevation and distance from the coast)".*

Response: Thank you for the suggestion. Your suggestion is also how we would explain the spatial variation in the length of the snow season, and is added to the revised manuscript.

*RC2: p8 line 4: The quality of the observations are fair . . . "Fair" is far too vague to be helpful – it might mean the results are reliable to within +/-5 or +/- 50%! Please provide a quantitative estimate of reliability.*
Response: A qualitative estimate is added to the revised manuscript. The uncertainty is assessed to be around 5% at 20 C and 6% at -20 C, for humidity in the form of vapour pressure (pers. comm. Gabriel Kielland, MET Norway).

*RC2: p8 line 7: red and purple. There is no good reason to use two colours for LW V sites in Fig 1g. See the notes on Figure changes below – I suggest using one colour that can easily be seen such as orange.* Response: This has been updated based on you suggestions below.

*RC2: p9 line 9: conducted this study > conducted for this study*
Response: Corrected.

*RC2: p9 line 14: time-series > time series [two words (e.g. see primary mathematical literature) only hyphenate when writing time-series analysis]*
Response: This has been corrected.

*RC2: p10 line 9: located at the ground > located on the ground* Response: Corrected.

*RC2: p10 line 13: time-series > time series* Response: Corrected

*RC2: p10 line 18: to not introduce spatial or temporal smoothing > to avoid introducing spatial or temporal smoothing*
Response: Thank you for the suggestion, the sentence is now updated accordingly.

*RC2: p10 line 23: analysed multiple > analysed using multiple* Response: Corrected.

*RC2: p11 lines 11 to 14: Use separate lines for each equation. Indicate that "mu" = mean.*
Response: The metrics are now included in an enumerated list, where "mu" is defined.

*RC2: p11 line 15: time-series > time series* Response: Ok

*RC2: p11 line 18: probability of the that the > probability that the*
Response: Corrected.

*RC2: p11 line 24: differs significantly > differ significantly [dependencies and model estimates are both plural in this sentence]*
Response: Corrected

*RC2: p12 line 2: a air mass > an air mass*
Response: Corrected

*RC2: p12 line 16: time-series > time series*
Response: Corrected

*RC2: P13 line 19: weakened with 0.11 hPa/100 m > weakened by 0.11 hPa/100 m*
Response: Corrected

*RC2: p16 line 21: fall is > fall (autumn) is*
Response: Updated.

*RC2: p20 line 21: trough > through*
Response: Corrected. Thank you for noticing this.

*RC2: p22 line 22: higher found in > higher than found in*
Response: Corrected

*RC2: p23 line 11: average to a five times larger > average in a five times larger*
Response: Corrected

*RC2: p25 line 12: produced points to that great care > produced indicates that great care*
Response: Updated

*RC2: P25 line 23: sensitivity too > sensitivity to*
Response: Corrected

*RC2: p27 line 9: conducting in the production > conducted in the production*
Response: Corrected

*RC2: p27 line 12: which might to do > which might be to do*
Response: Updated to "trend. This might be due to the"

*RC2: p27 line 16: does not implies > does not imply*
Response: Corrected

*RC2: p28 line 26: e.g. in ISI-MIP. Provide reference to ISI-MIP research*
Response: The revised manuscript now included a reference to *Teklesadik et al. 2017*, where the impact of climate change on surface hydrology on in the Upper Blue Nile basin is examined using bias corrected global climate model data constructed using the ISI-MIP trend preserving bias-correction approach (*Hempel et al. 2013*) as forcing data for hydrological models.

*RC2: p28 line 27: made available may easily > made available and may easily*

Response: Corrected

**RC2: Figure changes:**

*RC2: Fig. 1f and 1g The resolution of the background map plus the background colours (green, yellow, white) make it very hard to see the plotted points in blue and purple. I suggest removing the background colour in 1f and plot points in red. There is no need to plot the LW V stations in different colours in 1g. Use a colour that stands out instead (e.g. orange for LW V sites).*
Response: Thank you for these suggestion. Plot 1f and 1g no longer shows terrain in the background, and the VP stations are now marked with red, and the LW stations are marked with orange. An additional update was made; the coastline is no longer plotted in plot a)-e), to make the differences in the  land-sea masks more visible.

*RC2: Fig 2 Make sure the observations stand out. I suggest using a thick continuous black line instead of a thin dashed line lacking plotted points. Increase the size of all the text (including key, heading and axis label) relative to the figure. Add a horizontal axis label "Calendar month".*
Response: Corrected

*RC2: Fig 3. It is extremely difficult to distinguish these lines. Avoid using red and green on the same plot. Remove all the plotted shapes – it confuses things. I suggest instead using a thick continuous line to make the observations easily distinguished. For the models use fewer colours according to dataset type (e.g. MERRA and ERA-Interim = red; PGF and WFDEI = blue; V1 and V2 = orange, HySN = grey). Then distinguish within pairs of lines of the same dataset type by using one continuous and one dashed (e.g. MERRA continuous red, ERA-Interim dashed red). Add explanation of the key letters to the caption.*
Response: The figure has now been updated with a thick continuous line for the observations and the remaining datasets are grouped as suggested according to data source. The letter keys are explained in the title.

*RC2: Fig. 5 Make plotted symbols much bigger.*
Response: The symbols have been enlarged and their colors have been updated to be consistent with the color changes as outlined above.

*RC2: Fig 6. Use system suggested for fig. 3.*
Response: Assuming this refers to Fig 7. Fig. 7 is updated with the system used in fig. 3

*graham.weedon@metoffice.gov.uk 31st Jan 2019*

References added

[revised manuscript text omitted]